# Low-cost eddy covariance: a case study of evapotranspiration over agroforestry in Germany

Christian Markwitz[1] and Lukas Siebicke[1]

[1]Bioclimatology, University of Goettingen, Büsgenweg 2, 37077 Göttingen, Germany

*Correspondence to:* Christian Markwitz (christian.markwitz@forst.uni-goettingen.de)

**Abstract.** Heterogeneous land surfaces require multiple measurement units for spatially adequate sampling and representative fluxes. The complexity and cost of traditional eddy covariance set-ups typically limits the feasible number of sampling units. Therefore, new low-cost eddy covariance systems provide ideal opportunities for spatially replicated sampling.

The aim of this study was to test the performance of a compact low-cost pressure, temperature and relative humidity sensor for the application of evapotranspiration measurements by eddy covariance over agroforestry and conventional agriculture in Germany. We performed continuous low-cost eddy covariance measurements over agroforestry and conventional agriculture for reference, at five sites across Northern Germany over a period of two years from 2016 to 2017. We conducted side-by-side measurements using a roving enclosed-path eddy covariance set-up to assess the performance of the low-cost eddy covariance set-up.

Evapotranspiration measured with low-cost eddy covariance compared well with fluxes from conventional eddy covariance. The slopes of linear regressions for evapotranspiration comparing low-cost and conventional eddy covariance set-ups ranged from 0.86 to 1.08 for five out of ten sites, indicating a 14% flux underestimation and a 8 % flux overestimation relative to the conventional EC set-up, respectively. Corresponding coefficients of determination, $R^2$, ranged from 0.71 to 0.94 across sites. The root mean square error for differences between latent heat fluxes obtained by both set-ups were small compared to the overall flux magnitude, with a mean and standard deviation of $34.23 \pm 3.2 \, \mathrm{W \, m^{-2}}$, respectively, across sites.

The spectral response characteristics of the low-cost eddy covariance set-up were inferior to the eddy covariance set-up in the inertial sub-range of the turbulent spectrum. The water vapour flux cospectrum of the low-cost eddy covariance set-up underestimated the theoretical slope of -4/3 stronger than the conventional eddy covariance set-up. This underestimation was mainly caused by the limited response time of the low-cost thermohygrometer longer than one second.

We conclude that low-cost eddy covariance sensors are an alternative to conventional eddy covariance sensors when, first, replicates are required and, second, the spatial variability of fluxes of the ecosystems of interest is larger than above reported set-up specific differences in fluxes.

# 1 Introduction

Eddy covariance (EC) is often the method of choice for measurements of the ecosystem-atmosphere exchange of water vapour, sensible heat, momentum and trace gases (Baldocchi (2003), Baldocchi (2014), Farahani et al. (2007)) over a variety of ecosystems. At ecosystems with spatial variability of surface cover, the representativity of the measured fluxes is limited by the flux footprint extend (Schmid, 2002). Either the spatial variability of fluxes remains undetected (for small footprints) or can not be resolved explicitly (for large footprints). Such heterogeneous ecosystems require multiple towers for spatially representative flux sampling.

While the single-tower approach is still most common for ecosystem studies, a few studies have performed replicated EC measurements. Davis et al. (2010) studied carbon fluxes over an arable site in South East Ireland. Loescher et al. (2017) used a set of two flux towers separated by a distance of 775 m for uncertainty estimation of EC flux measurements.

Replication of sampling points was traditionally limited by high costs and the complexity of conventional EC set-ups. Therefore, there is increasing interest in the development of low-cost sensors for different applications in the biogeosciences.

Dias et al. (2007) proposed a cost-efficient direct attenuated EC set-up to measure latent heat fluxes, combining a sonic anemometer and a hygrometer of fast response. They applied a correction factor to the time-domain covariance between the vertical velocity and relative humidity measurements. Hill et al. (2017) presented a low-cost measuring set-up to measure both $CO_2$ and water vapour fluxes and discussed the value of increasing the number of measuring complexes for the statistical power of EC measurements in a variety of landscapes. Hill et al. (2017) concluded that at least four flux towers per site are required to confirm a statistical confidence of $95\%$ that the flux over one year is not zero and therefore accept to a statistical confidence of $5\%$ that the annual flux is zero. This is of major importance for an ecosystem, which is heterogeneous at a scale larger than the flux footprint of a single tower.

Besides the replication of measurement units within one ecosystem, the ecosystem-to-ecosystem replication of sampling points is of importance to e.g. assess the potential of forests for climate change mitigation and as a $CO_2$ sink (De Stefano and Jacobson, 2018). The outcome of synthesis studies, e.g. on the water use of terrestrial ecosystems at global scale (Tang et al., 2014) could be strengthened by an increased number of flux measuring units across ecosystems. Low-cost instrumentation can foster replicated EC measurements across the globe, especially in ecoregions that are currently only sparsely sampled, such as Africa, Oceania (except Australia) and South America (Hill et al. (2017) and Table 1 therein). With replicated measurements of low cost, effects of land-use changes or different agriculture management practices on turbulent fluxes can be assessed. A prominent example are flux measurements over heterogeneous shaped short rotation alley cropping systems (ACS) as one type of agroforestry (AF) in comparison to monocultural agriculture systems. Flux measurements over AF require replicated measurements to capture the spatial variability of the turbulent fluxes both at a single AF system and across multiple AF systems.

Our objectives are (a) to test the performance of a new EC measuring complex under field conditions to measure half-hourly evapotranspiration over alley cropping agroforestry systems and monocultural agriculture systems, and (b) to evaluate the low-cost measuring complex relative to conventional EC instrumentation.

## 2 Material and Methods

### 2.1 Site description

The study is part of the SIGNAL (Sustainable intensification of Agriculture through agroforestry) project (http://www.signal. uni-goettingen.de/), which aims to evaluate the sustainability of agroforestry in Germany. It is based on data collected at five sites in Northern Germany (Fig. 1). Each site consists of an agroforestry (AF)- and a monocultural control plot (MC). The agroforestry plots are alley cropping systems, consisting of fast growing trees, such as willow [*Salix*], poplar [*Populus*] and black locust [*Robinia*], interleaved by either annually rotating crops or perennial grassland. The control plots consist of the same crop or grass type as planted between the tree strips and are managed as monocultural agriculture. Three sites undergo annual crop rotation (Dornburg, Forst and Wendhausen), while two systems are of a perennial grassland type (Mariensee and Reiffenhausen). The project design includes a fixed tree alley width of 10 m, while alley length and number are variable across sites. Tree alley distances vary between 10 m, 24 m, 48 m and 96 m. The area covered by trees in relation to the whole agroforestry plot area varies between 6 % and 72 %. Table A1 provides an overview of site locations, agroforestry geometry and stand characteristics.

We performed a flux footprint climatology analyses with the flux footprint prediction online tool (http://footprint.kljun.net/, Kljun et al. (2015)). The flux footprint climatology is valid for the respective campaign and only for daytime data according to a global radiation $R_G > 20\,W\,m^{-2}$. We found a 90 % flux magnitude contribution of the agroforestry plot of Forst and the monoculture plot of Dornburg and a 80% flux magnitude contribution of the agroforestry plots of Dornburg and Wendhausen. The smallest agroforestry system of Reiffenhausen contributed the least to the measured turbulent flux with 60 %. Outside the agroforestry plot, fluxes were affected by nearby crop fields in about 400 m distance to the flux tower in northerly direction and forest in about 200 m distance in southerly direction.

### 2.2 Instrumental set-up

#### 2.2.1 Standard meteorological measurements

Continuous measurements of micrometeorological and standard meteorological variables were performed since March 2016. At each agforestry plot one eddy covariance mast with a height of 10 m was installed and at each monocultural plot one eddy covariance mast with a height of 3.5 m. Each mast at the agroforestry and the monocultural plot was equipped with an identical instrumental set-up. An overview of all installed instruments is given in Table 1. The data were logged and stored on a CR6 data logger (*Campbell Scientific*, Inc., Logan, UT, USA). The meteorological data were regularly sent to a database via mobile phone network.

#### 2.2.2 Conventional eddy covariance installation

Fluxes of sensible heat and momentum were continuously measured with a uSONIC3-omni (*METEK GmbH*, Elmshorn, Germany) ultrasonic anemometer. $CO_2$ and water vapour fluxes were measured in campaigns during the vegetation periods of

**Table 1.** Instrumentation for flux and meteorological measurements used at all five agroforestry and five monocultural agriculture plots.

| Variable | Height [m] | Instrument | Company |
|---|---|---|---|
| Standard meteorological measurements | | | |
| 3D wind components, u, v, w, sonic temperature, $T_s$, wind speed and -direction | 3.5,10 | uSONIC-3 Omni | METEK GmbH, Elmshorn, Germany |
| Net radiation, $R_N$ | 3, 9.5 | NR-Lite2 Net Radiometer | Kipp&Zonen, Delft, The Netherlands |
| Global radiation, $R_G$ | 3, 9.5 | CMP3 Pyranometer | Kipp&Zonen, Delft, The Netherlands |
| Relative humidity, RH, air temperature, T | 2 | Hygro-Thermo Transmitter-compact (Model 1.1005.54.160) | Thies Clima, Göttingen, Germany |
| Precipitation | 1 | Precipitation Transmitter (Model 5.4032.35.007) | Thies Clima, Göttingen, Germany |
| Atmospheric pressure, ppp | 0.5 | Baro Transmitter (Model 3.1157.10.000) | Thies Clima, Göttingen, Germany |
| Ground heat flux, G | -0.05 | Hukseflux HFP01 | Hukseflux, Delft, The Netherlands |
| Soil temperature, $T_{Soil}$ | -0.02, -0.05, -0.10, -0.25, -0.5 | DS18B20 | |
| Conventional eddy covariance measurements | | | |
| u, v, w, $T_s$ | 3.5,10 | uSONIC-3 Omni | METEK GmbH, Elmshorn, Germany |
| Water vapour mole fraction, $C_{H_2O_v}$ | 3.5, 10 | LI-7200 | LI-COR Inc., Lincoln, Nebraska (USA) |
| Carbon dioxide mole fraction, $C_{CO_2}$ | 3.5, 10 | LI-7200 | LI-COR Inc., Lincoln, Nebraska (USA) |
| Low-cost eddy-covariance measurements | | | |
| u, v, w, $T_s$ | 3.5,10 | uSONIC-3 Omni | METEK GmbH, Elmshorn, Germany |
| RH, T, ppp | 3,9.5 | BME280 | Robert Bosch GmbH, Stuttgart, Germany |

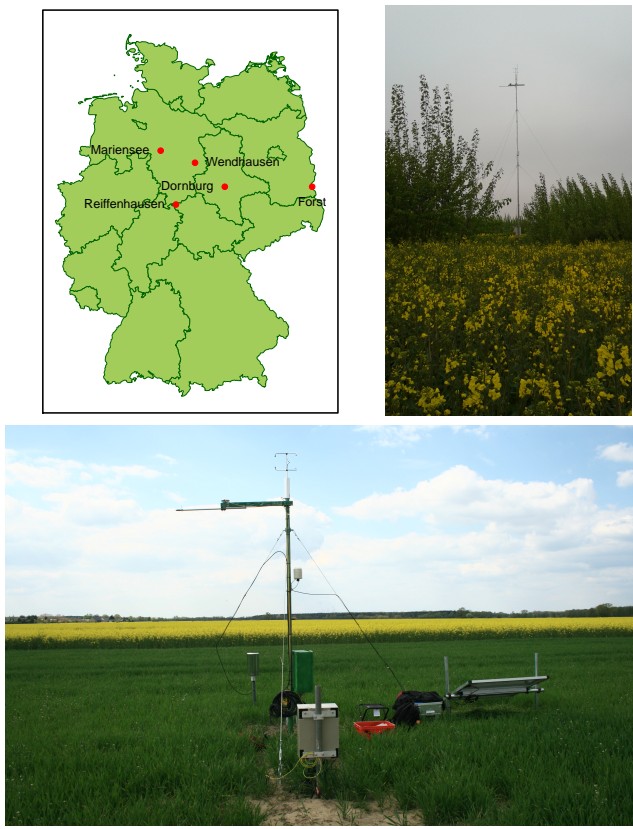

**Figure 1.** Top left: SIGNAL sites (Map source: Bundesamt für Kartographie und Geodäsie (2011)); top right: agroforestry plot in Dornburg with eddy covariance mast; bottom: monocultural agriculture plot in Forst (Lower Lusatia) with eddy covariance mast.

2016 and 2017. During the 2016 campaign, fluxes were measured during two consecutive periods of four weeks duration separately at the agroforestry and monocultural plots, whilst in 2017 both plots were sampled simultaneously over a time period of approximately four weeks (see Table A2 for exact dates). During the campaigns, the instrumentation specified in Table 1 was complemented by a LI-7200 (*LI-COR Inc.*, Lincoln, Nebraska, USA) enclosed-path infrared gas analyser (Burba et al., 2012).

5 The data were measured together with the three-dimensional wind velocity and the sonic temperature and stored on the same data logger (CR6 , *Campbell Scientific Ltd.*, Bremen, Germany) as used for the meteorological variables. The water vapour and $CO_2$ mole fractions were sampled with a sampling frequency of 20 Hz. The intake tube was of 1 m length and had an inner tube diameter of 5.3 mm (2016) and 8.2 mm (2017). The separation of the gas analysers intake tube relative to the centre of the sonic anemometer was different for each plot and is summarized in Table A3. The flow rate was kept constant at 15 slpm.

### 2.2.3 Low-cost eddy-covariance (EC-LC) installation

The low-cost eddy-covariance set-up shared the same ultrasonic anemometer (uSONIC3-omni) as used for the conventional EC set-up. The water vapour mole fraction was derived from the combined digital pressure, relative humidity and air temperature sensor BME280 manufactured by *Robert Bosch GmbH*, Stuttgart, Germany (hereafter named thermohygrometer). Figure 2 depicts the low-cost set-up. The measuring principle of the air pressure sensor is resistive, for the relative humidity sensor capacitive and for the temperature sensor is based on diode voltage measurements. The ultrasonic anemometer measured the three-dimensional wind speed and the ultrasonic temperature at a frequency of 20 Hz, whereas the thermohygrometer measured the air temperature, relative humidity and air pressure at a sampling frequency of 8 Hz. The specified response time of the thermohygrometer for relative humidity measurements is 1 s to overcome 63% of a step change from 90% to 0% or 0% to 90% relative humidity at 25 °C air temperature.

The response time of the temperature sensor of the thermohygrometer was not explicitly stated. Therefore, we estimated the response time in a lab experiment. We exposed the temperature sensor to a rapid temperature change about 10 °C warmer than ambient air temperature. The time constant $\tau$ was then directly proportional to the slope of the linear regression fit

$$t = \tau \ln \left( \frac{\vartheta(t=1) - \vartheta_{Ambient}}{\vartheta(t=t_{var}) - \vartheta_{Ambient}} \right)$$

with the measurement time, $t$, the air temperature at the first time step, $\vartheta(t=1)$, the ambient air temperature, $\vartheta_{Ambient}$, and air temperature at variable time step, $\vartheta(t=t_{var})$. The time constant achieved for the temperature sensor was 23.3±0.9 s as a mean of four replications. During the lab experiment the thermohygrometer was placed inside the same housing as deployed in the field.

The thermohygrometer was placed 0.5 m below the centre of the sonic anemometer in a PVC housing to protect the thermohygrometer from precipitation. The PVC housing consisted of an outer and an inner cylinder. The inner cylinder was perforated on the top to provide a continuous air flow of 15 lpm, generated by a ventilator (HA30101V3-0000-A99, *Sunonwealth Electric Machine Industry Co. Ltd.*, Fresnes Cedex, France). The ventilator was placed below the thermohygrometer inside the inner cylinder. The volume of the inner cylinder was 98.1 cm$^3$.

The absolute accuracy tolerance of the relative humidity sensor was specified as $\pm 3$ % for 20 % to 80 % relative humidity at 25 °C air temperature. For the temperature sensor an absolute accuracy tolerance of $\pm 0.5$ °C at 25 °C air temperature was given and for a temperature range of 0 °C to 65 °C an absolute accuracy tolerance of $\pm 1$ °C was specified. The pressure sensor has an absolute accuracy tolerance of $\pm 1$ hPa for a pressure range from 300 hPa to 1100 hPa at air temperature between 0 °C and 65 °C (Bosch Sensortec GmbH, 2016).

Digital data from the thermohygrometer were recorded via the i2c protocol and stored on a RaspberryPi model B+ (*Raspberry Pi Foundation*, Cambridge, UK). The thermohygrometer has very low power consumption of approximately 3.6 $\mu A$ at a sampling frequency of 1 Hz. The power draw of the thermohygrometer is 9.4e-5 W at a measuring frequency of 8 Hz, if powered with 3.3 V and if all three variables are measured at the same time. The RaspberryPi has a maximum power consumption of about 1.1 W.

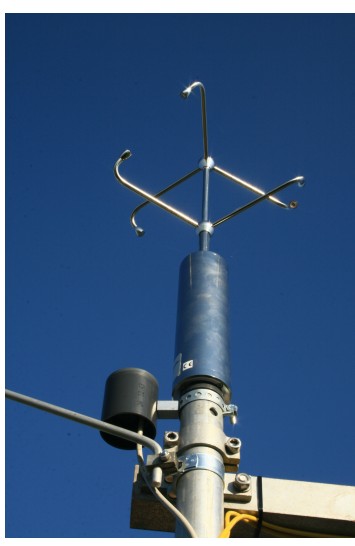

**Figure 2.** Low-cost eddy covariance instrumentation, featuring a uSONIC3-omni sonic anemometer and a BME280 thermohygrometer. The thermohygrometer is placed in a ventilated PVC housing below the sonic anemometer.

The potential of the low-cost EC set-up are replicated measurements of evapotranspiration across different ecosystems. The relative cost of the low-cost set-up (featuring a sonic anemometer, a RaspberryPi and the thermohygrometer of low cost) is often less then $10\%$ of a typical conventional EC set-up. Beside a precipitation protection and a stable power supply, the thermohygrometer needs low maintenance. The mean time before failure of the sensor in our study was approximately 2 years.

5  ## 2.3  Flux computation

### 2.3.1  Conventional eddy covariance set-up

Latent heat fluxes and sensible heat fluxes were calculated with the open source EddyPro® eddy covariance software (*LI-COR, Inc.*, Lincoln, Nebraska, USA, version 6.2.0).

The fluxes were computed as

$$H = \rho_a c_p \overline{w' T_s'} \tag{1}$$

$$\lambda E_{EC} = \lambda M_{H_2O_v} \overline{w' d'_{H_2O_v}} \tag{2}$$

with the density of dry air, $\rho_a$, the specific heat capacity at constant pressure, $c_p$, the vertical velocity component, $w$, the ultrasonic temperature, $T_s$, the latent heat of evaporation, $\lambda$, the molar mass of water vapour, $M_{H_2O_v}$, and the molar density of water vapour, $d_{H_2O_v}$. Primes denote deviations from the mean and overlines denote time averages.

15  Fluxes were calculated over a block averaging period of 30 minutes. The horizontal wind component was rotated into the mean wind direction via double rotation (Kaimal and Finnigan, 1994). Time lags between the ultrasonic anemometer and the intake tube of the LI-7200 gas analyser were calculated and corrected as a function of relative humidity (LI-COR, 2015).

The effect of density fluctuations on the turbulent fluxes was corrected for by the WPL correction (Webb et al., 1980) and the ultrasonic temperature was corrected for humidity effects (Schotanus et al., 1983). Fluxes of sensible and latent heat, and momentum were filtered by removing all flux values corresponding to a flag of 2, following the two-stage quality control procedure of Mauder and Foken (2011b). Latent heat fluxes below -50 W m$^{-2}$ and above 500 W m$^{-2}$ were discarded. We further discarded latent heat fluxes according to the 97.5 % percentile of the $H_2O$ variance and spikes were removed after Vickers and Mahrt (1997). Through quality check $9.6 \pm 3.2$ % of half-hourly latent heat fluxes obtained by the EC set-up were discarded and $10.4 \pm 3.8$ % of half-hourly latent heat fluxes obtained by the EC-LC set-up were discarded, as a mean over all five plots. Low-frequency and high-frequency losses were corrected by the procedure of Moncrieff et al. (2004) and Ibrom et al. (2007), respectively. Random uncertainties of fluxes were calculated after Mann and Lenschow (1994).

## 2.3.2 Low-cost eddy covariance set-up

The latent heat flux from the low-cost eddy covariance set-ups was calculated as the covariance between the vertical velocity and the water vapour mole fraction, again with the EddyPro® eddy covariance software (*LI-COR, Inc.*, Lincoln, Nebraska, USA, version 6.2.0). The water vapour mole fraction, $C_{H_2O_v}$, was derived from relative humidity, temperature and pressure measured with the thermohygrometer from the definition of the specific humidity, $q$, as the quantity of water vapour per quantity of moist air. The latter two quantities were expressed as the density of water vapour, $\rho_{H_2O_v}$, and moist air, $\rho_m$, respectively. The density of moist air is defined as the sum of the density of dry air, $\rho_d$, and the density of water vapour.

$$q = \frac{\rho_{H_2O_v}}{\rho_m}$$
$$= \frac{\rho_{H_2O_v}}{\rho_d + \rho_{H_2O_v}} \tag{3}$$

We then replaced the density of water vapour and the density of dry air in Eq. (3) as per Eqs. (4) and (5), respectively,

$$\rho_{H_2O_v} = \frac{C_{H_2O_v} \cdot M_{H_2O_v}}{V_m} \tag{4}$$

$$\rho_d = \frac{p - e}{R_d \cdot T_A} \tag{5}$$

with the molar mass of water vapour, $M_{H_2O_v} = 18.02$ g mol$^{-1}$, the molar volume of air

$$V_m = \frac{\Re \cdot T_A}{p} \, (m^3 \, mol^{-1}), \tag{6}$$

the universal gas constant, $\Re = 8.314$ J mol$^{-1}$K$^{-1}$, and the specific gas constant of dry air, $R_d = 287.058$ J kg$^{-1}$ K$^{-1}$ .

Solving Eq. (3) for $C_{H_2O_v}$ leads to the water vapour mole fraction

$$C_{H_2O_v} = \frac{q \Re (p - e)}{p M_{H_2O} R_d (1 - q)}. \tag{7}$$

The specific humidity in Eq. (7) was calculated as a function of relative humidity, temperature and air pressure measurements from the thermohygrometer:

$$q = 0.622 \cdot \frac{e}{p} \tag{8}$$

The saturation vapour pressure, $E_{Sat}$, and vapour pressure, $e$, in Eq. (8) were calculated using Eqs. (11) and (12), respectively.

The water vapour mole fraction is expressed as the wet mole fraction, thus the mass of water vapour molecules per total mass of air. Therefore, latent heat fluxes derived from the water vapour mole fraction needs to be corrected for density effects (WPL correction, Webb et al. (1980)) caused by temperature and water vapour fluctuations. The WPL correction requires true ambient air temperature measurements. Our fast measurements of the true air temperature obtained by the thermohygrometer were attenuated by the slow response time of the thermohygrometers temperature measurements. Additionally, the air temperature obtained by the thermohygrometer overestimated the ultrasonic temperature used as a reference caused by a radiation effect from the grey PVC housing. Therefore, we derived a true air temperature for the WPL correction from the definition of the ultrasonic temperature, $T_s$, and its dependency on air humidity

$$T_s = T \left(1 + 0.32\frac{e}{p}\right) \tag{9}$$

with the atmospheric pressure, $p$, to calculate a moisture corrected temperature, which we used as an estimate of true air temperature, $T$:

$$T = \frac{T_s}{\left(1 + 0.32\frac{e}{p}\right)} \tag{10}$$

An initial value for the vapour pressure in Eq. (10) was calculated from an approximation of the saturation vapour pressure, $E_{Sat}$ (based on $T_s$) (Stull, 1989) and from relative humidity, *RH*,

$$E_{Sat} = 0.6112 \exp \frac{17.6294 \cdot (T_s - 273.16)}{T_s - 35.86K} \tag{11}$$

$$e = \frac{RH \cdot E_{Sat}}{100} \tag{12}$$

The derivation of the vapour pressure was iterated using Eqs. (9), (10), and (11).

We matched the water vapour mole fraction calculated from the thermohygrometer data and the velocity components measured with the ultrasonic anemometer according to the nearest-neighbour date values to address the two different sampling frequencies of 8 Hz and 20 Hz, respectively. The two data acquisition systems (the CR6 logger and the RaspberryPi, respectively) were regularly manually synchronized. In detail, the RaspberryPi was synchronized with an online ntp server, whereas the CR6 logger was synchronized during regular maintenance visits.

A timelag between the anemometer and the thermohygrometer was corrected for in a preprocessing routine. The cross-correlation function *ccf* from the R-package *tseries* (Trapletti and Hornik, 2017) was used to detect the timelag between the vertical velocity component and the water vapour mole fraction. The respective timelag was extracted according to the maximum cross-correlation coefficient. The estimated lag time was used to merge the velocity components, u, v, w, and the ultrasonic temperature with the nearest-neighbour water vapour mole fraction.

We applied the same flux corrections and quality checks to fluxes obtained by the EC-LC set-up as for the conventional EC set-up (see Sect. 2.3.1). The only difference was the correction of high-frequency losses, where we applied the correction after

Moncrieff et al. (1997). The correction procedure was explicitly recommended by Moncrieff et al. (1997) for either open-path sensors or closed-path systems of very short and heated sampling lines.

The method is fully analytic and for each half-hour period the flux cospectra are estimated from analytical formulations after Moncrieff et al. (1997) (Eqs. (12)-(18) therein). Those equations are a modified version of the formulas in Kaimal et al. (1972). The cospectra are expressed as a function of the normalized frequency, which is a function of the natural frequency, measurement height, zero displacement height, wind speed and atmospheric stability.

We studied the impact of the different corrections on the raw turbulent evapotranspiration rates, obtained by the EC-LC set-up. We applied the single corrections separately on a test data-set from the agroforestry plot in Dornburg from the 14-July to the 12-August, 2016. We assessed the impact of the following corrections on the raw evapotranspiration rates: 1) the fully analytic high-frequency cospectral correction after Moncrieff et al. (1997), 2) the low-frequency cospectral correction after Moncrieff et al. (2004) and 3) the WPL correction after Webb et al. (1980). The corresponding results are presented in Sect. 3.3.

Linear regression analyses were performed between evapotranspiration obtained by the EC set-up and the EC-LC set-up. We used the major axis linear regression method from the *lmodel2* function as part of the lmodel2 R-package (Legendre and Oksanen, 2018). The major axis linear regression method assumes equally distributed errors in both time series.

## 2.4   Spectral analysis

Commonly, high-frequency trace gas measurements (e.g. the water vapour- or $CO_2$ mole fraction) taken by closed- or enclosed-path gas analysers are attenuated in the high-frequency range of the energy spectrum (Lenschow and Raupach, 1991). Attenuation is mainly caused by exchange processes (adsorption or desorption) of gas molecules with tubing walls (Leuning and Moncrieff (1990), Ibrom et al. (2007)). This effect is most severe for sticky gases such as water vapour. In contrast, the temperature spectrum and cospectrum is assumed to be not attenuated by the molecular exchange processes with tubing walls, as the measurements are taken with a sonic anemometer, which is open-path. Attenuation of the ultrasonic temperature and the wind velocity components is mainly caused by the path-averaging effect, especially at low wind speeds and at very high wavenumbers (Kristensen and Fritzjarrals, 1984), which is outside the inertial sub-range. Therefore, we quantified the frequency response characteristics of the EC- and EC-LC set-ups by ensemble averaged spectra and cospectra of water vapour fluxes and compared them with temperature spectra and cospectra.

Additionally, we followed the Kolmogorov law (Kolmogorov, 1991), which describes a theoretical energy decrease with increasing frequency in the inertial sub-range of $-5/3$. The same theory formulates an energy decrease of $-2/3$ for scalars and $-4/3$ for covariances in the inertial sub-range (Foken et al., 2004), if multiplied by the frequency. The inertial sub-range is the region of the spectrum where neither dissipation nor the generation of turbulent kinetic energy is important for the respective eddy. The eddies in the inertial sub-range receive energy from larger eddies and pass it on to smaller eddies (Stull, 1989). The corresponding results are presented in Sect. 3.5.

The spectral response characteristics of the LI-7200 gas analyser and the low-cost thermohygrometer were further investigated in terms of the cut-off frequency, $f_c$, derived from true water vapour spectra. We estimated the cut-off frequency as

the frequency of the intercept between the maximum water vapour spectral energy and the linear fit of the energy spectrum in the inertial sub-range (between 0.1 and 1 Hz) on a double logarithmic scale (see Fig. 3 for clarification). From the cut-off frequency we estimated the sensors time constant, $\tau_c$, with the following relationship

$$\tau_c = 1/(2\pi f_c) \tag{13}$$

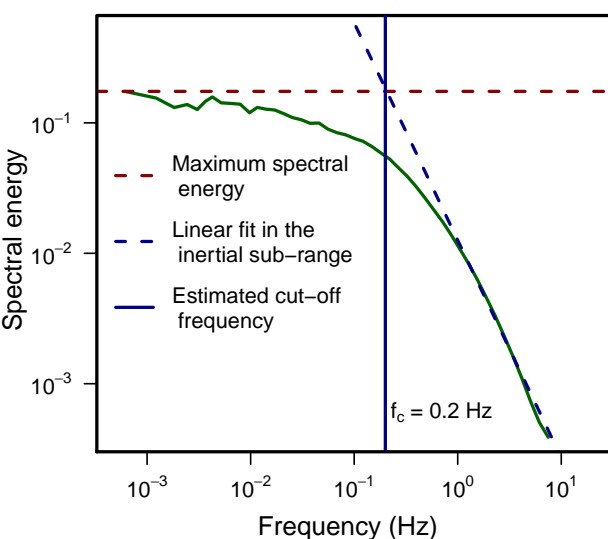

**Figure 3.** Sketch of the cut-off frequency estimation procedure with an exemplary true water vapour spectrum against frequency.

## 3 Results and discussion

### 3.1 Meteorological conditions

The measuring period at the monocultural agriculture plot of Dornburg (16-June to 14-July 2016) was characterized by high air temperature with a maximum daily mean of 25 °C and an average over the whole period of 18 °C (Fig. 4 (a) and Table A4). Cumulative precipitation over the period was low, with only 2 mm (Fig. 4 (a)). The low amount of rainfall caused a rapid ripening of the crops, which had a significant impact on the turbulent fluxes: evapotranspiration decreased and the sensible heat fluxes increased during the measuring period of four weeks.

In contrast, the measuring period (14-July to 12-August 2016) at the agroforestry plot in Dornburg (Fig. 4 (b)), about 500 m apart from the monocultural plot, was characterized by warm (mean air temperature of 19 °C) and humid ambient conditions with a cumulative precipitation of about 50 mm and a mean vapour pressure deficit (VPD) of 6.41 hPa. At the time of installation of the EC set-up the crops were already mature whilst the trees were at the seasonal maximum of their productivity.

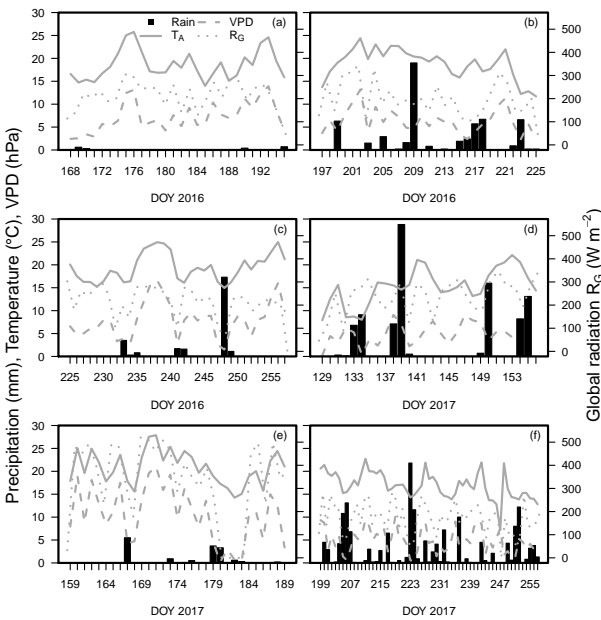

**Figure 4.** Daily averaged air temperature, vapour pressure deficit (VPD), daily summed precipitation and averaged global radiation, R$_G$, for the following plots at each subfigure: Dornburg monoculture, (a), Dornburg agroforestry, (b), Reiffenhausen agroforestry, (c), Wendhausen, (d), Forst, (e) and Mariensee, (f). For Wendhausen, Forst and Mariensee, we took the average between the agroforestry and monocultural plot to provide a general overview of the meteorological conditions during the campaign. The averaging was done because both plots at the three sites were sampled simultaneously and the distance between both plots was maximum 600 m. We assumed similar weather conditions.

The weather conditions during the measuring period at the agroforestry plot in Reiffenhausen (12-August to 14-September 2016, Fig. 4 (c)) were warm with mean daily air temperatures above 15 °C and a total mean of 19.31 °C. The period was characterized by a few intense precipitation events with a cumulative sum of 26.3 mm (Table A4) and a mean VPD of 8.02 hPa.

The following measuring campaign in Wendhausen (03-May to 02-June 2017) was characterized by low mean VPD values
5   of 5.4 hPa at the agforestry plot and 5.2 hPa at the monocultural plot. At the beginning of the campaign, mean air temperature was at its lowest between 10 °C and 15 °C, whilst at the end air temperature was between 15 °C and 20 °C. The mean air temperature was 16.6 °C at the agforestry plot and 15.5 °C at the monocultural plot (Fig. 4 (d) and Table A4). Plants were very productive in terms of transpiration both at the agroforestry (trees and crops) and the monocultural (only crops) plots.

In contrast, the campaign period in Forst (08-June to 08-July 2017) was very warm with mean air temperature of 21.4 °C at
10  the agroforestry plot and 21.2 °C at the monocultural plot. High VPD values of around 12 hPa indicate dry ambient conditions.

### 3.2 Evapotranspiration rates from conventional- and low-cost eddy covariance

Diel cycles of evapotranspiration were well represented by the EC-LC set-up compared to the EC set-up on a 30-minute time scale (Fig. 5) at all sites. On a longer time scale (over a period of four weeks) the EC-LC set-up showed changes in daily

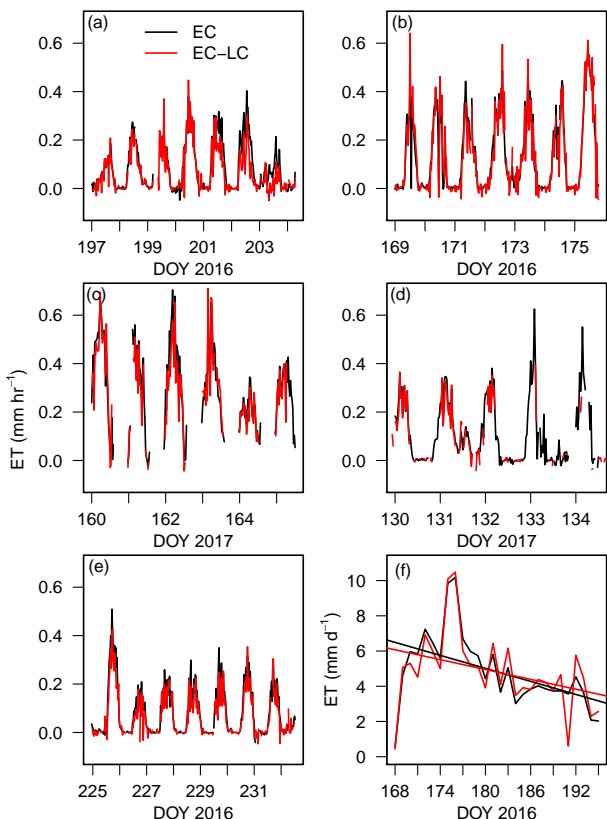

**Figure 5.** Half hourly evapotranspiration rates of one exemplary week, measured with the conventional EC- (black) and the EC-LC set-up (red) for Dornburg agroforestry, (a), Dornburg monoculture, (b), Forst agroforestry, (c), Wendhausen agroforestry, (d), Reiffenhausen agroforestry, (e). Subfigure (f) shows time series of daily summed evapotranspiration for the EC and EC-LC set-ups for Dornburg monoculture over the whole campaign period (from 16-June to 14-July 2016). We included the linear trend lines with a slope of -0.1232 $\text{mm d}^{-1}$ and a p-value of 0.009595 (black line) for the EC set-up and a slope of -0.09337 $\text{mm d}^{-1}$ with a p-value of 0.06549 (red line) for the EC-LC set-up.

summed evapotranspiration rates from higher sums ($\approx 6\,\text{mm d}^{-1}$) at the beginning and lower sums ($\approx 3\,\text{mm d}^{-1}$) at the end of the measuring period (from 16-June to 14-July 2016) at the monocultural agriculture plot of Dornburg in the same way as the EC set-up did (Fig. 5 (f)). We interpret this as a result of the ripening process of the crops. The ripening process was intensified by an exceptionally low cumulative precipitation of about 2 mm over the entire campaign period (Fig. 4 (a)) and a resulting low soil water content (not shown).

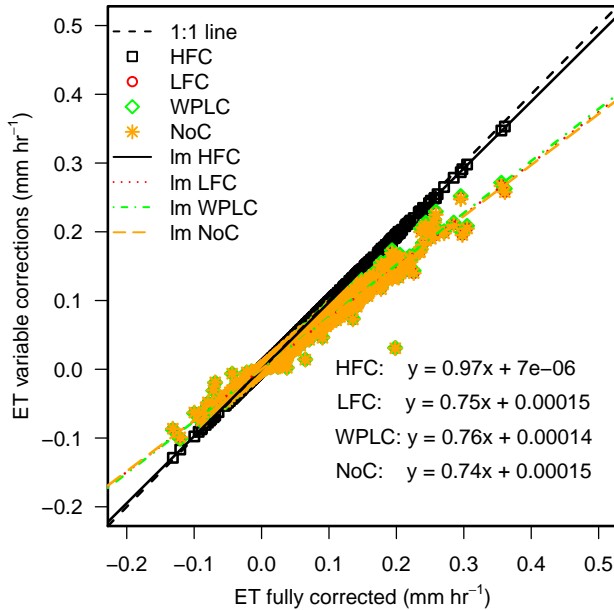

**Figure 6.** Evapotranspiration rates with the following corrections applied separately: 1) the high-frequency cospectral correction after Moncrieff et al. (1997) (HFC, black squares), 2) the low-frequency cospectral correction after Moncrieff et al. (2004) (LFC, red circles), 3) WPL correction after Webb et al. (1980) (WPLC, green diamonds) and 4) no correction (NoC, yellow stars) versus the fully corrected evapotranspiration rates of the EC-LC data set from Dornburg agroforestry. The best fit line with the same colours as the corresponding data points and the linear regression results for the respective corrections are shown. The linear regression is based on 1381 data points gathered during the campaign from the 14-July to 12-August 2016.

### 3.3 Effect of spectral- and WPL corrections on evapotranspiration rates from low-cost eddy covariance

A linear regression analysis between the uncorrected and the fully corrected evapotranspiration rates yielded a slope of 0.74 ($R^2$ = 99 %) (Fig. 6). The applied corrections accounted for an increase of 26 % of the overall flux magnitude.

5     The low-frequency cospectral correction after Moncrieff et al. (2004) accounted for 1 % of the fully corrected flux, which was the smallest contribution of all corrections to a flux magnitude increase.

The WPL correction yielded an increase of the flux magnitude of about 2 %. Other studies found an increase in the mean latent heat flux of 5.6 % (Mauder and Foken, 2006) when the WPL correction was applied. In the study of Mauder and Foken (2006), the WPL corrected latent heat flux measured with a LI-7500 open-path EC system was compared with an uncorrected flux from the same EC complex.

10     The high-frequency correction after Moncrieff et al. (1997) accounted for 23 % of the fully corrected flux, which was the largest contribution of all corrections to a flux magnitude increase. We interpret the high contribution of the correction from Moncrieff et al. (1997) as a result of the low response time of the thermohygrometer. In Ibrom et al. (2007) the low-pass

filtering properties of the closed-path system led to an underestimation of the measured latent heat flux and resulted in a necessary correction of 42 %.

The overall impact of spectral corrections on a change of the turbulent latent heat fluxes was stronger for the EC-LC set-up compared to the EC set-up. Here, we quantify the overall impact of spectral corrections on latent heat fluxes in terms of the spectral correction factor (SCF) calculated for each 30-minute period. The 30-minute SCF was multiplied with the respective uncorrected flux. A SCF larger than one indicates a flux magnitude increase, whereas a SCF lower than one indicates a flux magnitude decrease. Box-whisker plots of 30-minute SCFs for each site and each set-up are shown in Figure 7 (a). We found a mean SCF of $1.96 \pm 0.64$ for the EC-LC set-up and $1.14 \pm 0.05$ for the EC set-up across all sites, indicating a mean flux magnitude increase of 96 % for the EC-LC set-up and a mean flux magnitude increase of 14 % for the EC set-up. The mean SCF presented here integrates both night and day time periods. Thus, a high SCF during night time with commonly low latent heat fluxes leads to a smaller change of the flux magnitude than during day time, when fluxes are commonly high. Therefore, we also present the sum of 30-minute ET rates corrected for spectral losses and the sum of the total ET attributed to the spectral corrections in Figure 7 (b). The part of the total corrected ET attributed to the spectral corrections was higher for the EC-LC set-up compared to the EC-set-up and amounted on average to $42.7 \pm 14.1$ % of total ET for the EC-LC set-up and $9.3 \pm 3.3$ % of total ET for the EC set-up.

Across sites, we found the highest median spectral correction factor of 3.01 and the highest part of the total corrected ET attributed to the spectral corrections of 60.9 % for the EC-LC set-up at the monocultural agriculture plot of Dornburg. We interpret this as a measurement height dependency of the spectral corrections. The measurement height at the agroforestry plots was 10 m and at the monocultural agriculture plots the measurement height was 3.5 m. We assume that high-frequency eddies are more likely close to the surface. Therefore, a detected turbulent signal at the lower measurement height would be shifted towards high frequencies compared to the detected turbulent signal at the higher measurement height (Aubinet et al., 2012). If a sensor is not capable of detecting the turbulent signal in the high frequency range of the spectrum, the signal is attenuated and needs to be corrected.

### 3.4 Sensor cut-off frequency and time constant

The nominal time response of the relative humidity sensor as part of the thermohygrometer yields a theoretical sensor cut-off frequency of 0.16 Hz (6.3 s) calculated from Eq. 13.

Under field conditions we observed a mean cut-off frequency of $0.063 \pm 0.02$ Hz for the low-cost thermohygrometer and $0.3 \pm 0.2$ Hz for the LI-7200 gas analyser across five plots and all humidity classes (from 30 % to 90 % relative humidity bins). The respective mean time constant was $2.8 \pm 1$ s for the low-cost thermohygrometer and $0.6 \pm 0.3$ s for the LI-7200 gas analyser (see Fig. 8). For both sensors we found an exponential increase of the time constant with relative humidity (see Fig. 8).

Under field conditions, the cut-off frequency and the respective time constant of the thermohygrometer were inferior to the one given in the specifications. We interpret this as caused by the design of the enclosure. The thermohygrometer is placed at the end of a cylinder with the ventilator directly below, so that the flow velocity is decelerated. Subsequently, the decelerated

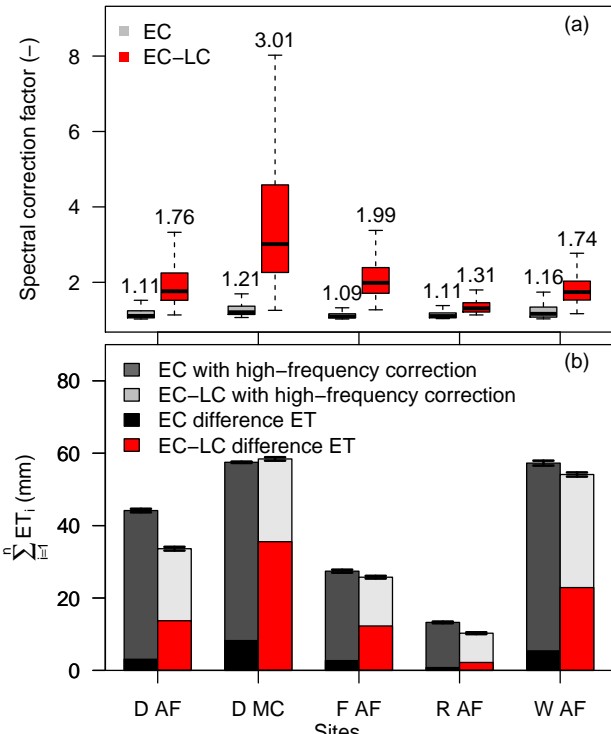

**Figure 7.** a) Box-whisker plot of spectral correction factors for the EC (grey) and the EC-LC (red) set-up for all sites. Values above the bars correspond to the median spectral correction factor, and, b) cumulative evapotranspiration rates for the EC and EC-LC set-ups for all sites, e.g Dornburg agroforestry, (D AF), Dornburg monoculture, (D MC), Forst agroforestry, (F AF), Wendhausen agroforestry, (W AF), and Reiffenhausen agroforestry, (R AF), over the respective campaign periods (Table A2). The error bars in Figure (b) correspond to the summed random uncertainties. The black and red bars correspond to that part of the total ET attributed to the high-frequency correction for the EC and EC-LC set-up, respectively. Incomplete records with either of EC or EC-LC missing were omitted.

flow velocity leads to a limited signal response. One suggestion for improvement of the frequency response would be to place the thermohygrometer inside a longer tube with a freely moving air stream. This ensures a faster air exchange inside the measurement cell of the thermohygrometer and hence a faster response time.

## 3.5  Spectral analysis

### 3.5.1  Ensemble averaged spectra of the water vapour mole fraction and sonic temperature and their dependency on relative humidity

The match of the water vapour mole fraction spectra with the theoretical -2/3 slope was found to be dependent on relative humidity. We observed the least deviation of the water vapour spectra obtained by the EC and EC-LC set-ups from the theoretical -2/3 slope for low relative humidity (Fig. 9). The relative humidity dependency of the water vapour spectra is a known feature

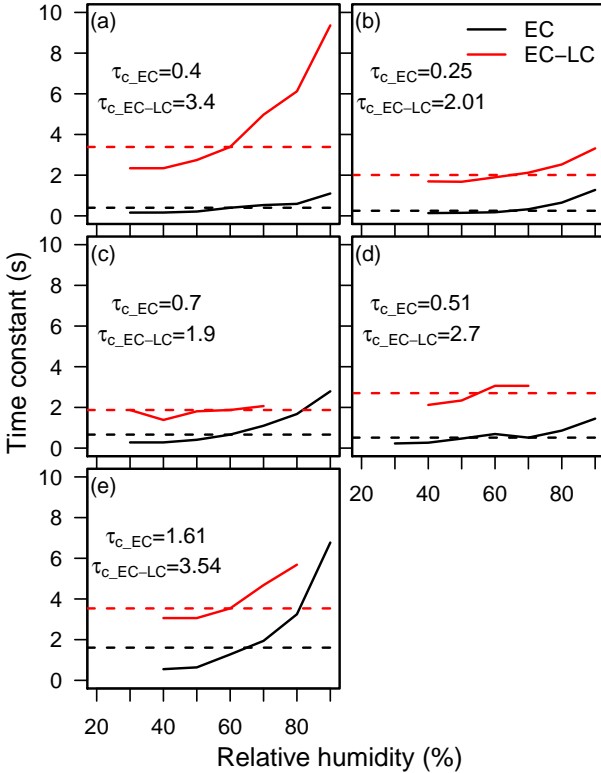

**Figure 8.** Time constant against relative humidity for the LI-7200 (black solid lines) and the thermohygrometer (red solid lines). Dashed lines with the same colour coding as for data shown and values written, correspond to the mean time constant for the respective sensors across all relative humidity classes. Sites correspond to Dornburg agroforestry, (a), Dornburg monoculture, (b), Forst agroforestry, (c), Reiffenhausen agroforestry, (d), and Wendhausen agroforestry, (e).

for closed- and enclosed-path gas analysers. Fratini et al. (2012) reported the same behaviour for both short (4 m) and very short (1 m) sampling lines. The so called "amplitude attenuation effect" (Fratini et al., 2012) was explained by Ibrom et al. (2007) as a result of absorption and desorption of water vapour molecules by hygroscopic particles inside the tube. Absorption and desorption processes are more pronounced at higher relative humidity and follow an exponential dependency on increasing
5    relative humidity (Fratini et al. (2012), Ibrom et al. (2007)).

The spectral response characteristics of the EC set-up were superior to the ones from the EC-LC set-up. The water vapour spectra from the EC-LC set-up deviated more from the theoretical -2/3 slope than the EC set-up in the inertial sub-range (between 0.1 Hz and 1 Hz) (Fig. 9). The ultrasonic temperature spectra followed a slope of -2/3 in the particular range of the energy spectrum, as the measurements are open-path.
10    For frequencies higher than 1 Hz, an increase of the spectral energy of water vapour for two out of five plots and both set-ups (i.e. Forst and Wendhausen agroforestry, Fig. 9 (c) and (d)) was observed, whereas the water vapour spectral energy increase for the agroforestry and monocultural plots of Dornburg and Reiffenhausen agroforestry was only found for the EC-LC set-up.

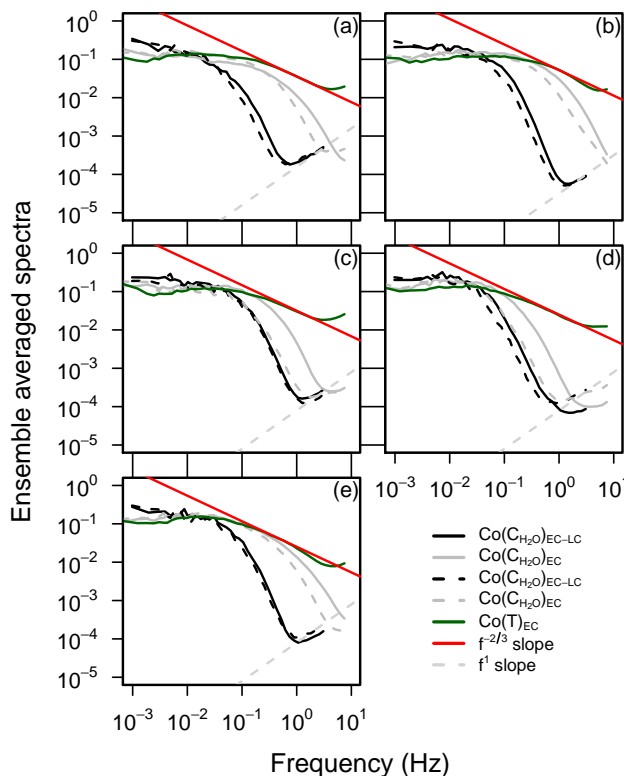

**Figure 9.** Ensemble averaged normalised water vapour and temperature spectra for relative humidity thresholds of 60 % (solid lines) and 80 % (dashed lines) versus the natural frequency. Spectra of the EC set-up (grey) and the EC-LC set-up (black) are shown. Subfigures correspond to plots: Dornburg agroforestry, (a), Dornburg monoculture, (b), Forst agroforestry, (c), Wendhausen agroforestry, (d), and Reiffenhausen agroforestry, (e). Spectra were filtered for low quality data, corresponding to a flag of 2 following the procedure of Mauder and Foken (2011a) and according to spike removal methods described in Vickers and Mahrt (1997). Relative humidity classes correspond to ancillary relative humidity measurements.

We interpret the spectral energy increase of water vapour in the particular frequency range as sensor noise, as indicated by the $f^1$ slope for white noise (Eugster and Plüss, 2010) in Fig. 9. The ultrasonic temperature spectra showed a slight spectral energy increase from frequencies higher than 4 to 5 Hz, which we interpret as an attenuation effect caused by the path-averaging (Kristensen and Fritzjarrals, 1984).

5    The observed noise of the water vapour spectra obtained by the EC set-up at the agroforestry plots of Forst and Wendhausen (Fig. 9 (c) and (d)) might be caused by different tube diameters in 2016 and 2017. In 2017 a thicker tube with an inner diameter of 8.2 mm was used compared to 2016 (inner tube diameter of 5.3 mm). In both years, a flow rate of 15 slpm was applied. The change in the inner tube diameter led to more turbulent conditions within the thinner tube than within the thicker tube. The thinner tube had a Reynolds number of 3950.6 (towards turbulent flow) and the thicker tube had a Reynolds number of 2551.71

10    (towards laminar flow).

### 3.5.2 Ensemble averaged cospectra of the water vapour flux and sensible heat flux

The water vapour flux cospectra deviated negatively from the theoretical -4/3 slope for the EC and EC-LC set-ups between a normalized frequency of 0.1 and 8 (the inertial sub-range) for all sites (Fig. 10). The deviation from the -4/3 slope in this particular frequency range was strongest for the EC-LC set-up, which is result of the limited spectral response characteristics of the thermohygrometer. As discussed in Section 3.4 the response time of the thermohygrometer was lower than given in the specifications.

The water vapour flux cospectra of the conventional EC set-up at the agroforestry plots of Forst and Wendhausen (Fig. 10 c) and d)) showed a stronger attenuation in the inertial sub-range, compared to the agroforestry plot and the monocultural agriculture plot of Dornburg and the agroforestry plot of Reiffenhausen (Fig. 10 a), b) and e)). That was likely caused by the different tube diameter at the respective plots and the effect on the turbulence characteristics inside the tubes, as discussed in Sect. 3.5.1.

At normalized frequencies higher than 8, we found a slope decrease of the water vapour flux cospectra obtained by the EC-LC set-up at all sites, which we interpret as an effect of sensor noise. Assuming that the vertical wind velocity measurements are unaffected by sensor noise, only the thermohygrometer measurements contribute to the slope decrease of the water vapour flux cospectra found in Fig. 10 for the EC-LC set-up.

In the low-frequency range (for a normalized frequency $< 0.1$) of the turbulent spectrum, the normalized water vapour cospectrum obtained by the EC-LC set-up was higher than the temperature cospectrum (Fig. 10). We interpret this finding as an effect of aliasing, which is an increased spectral energy in the low-frequency range due to a wrong representation of the high frequencies (Foken, 2008). That implies a too high sampling frequency relative to the sensors response time. The effect of aliasing was also observed for the EC cospectrum, but was much lower compared to the EC-LC set-up.

### 3.6 Water vapour molar densities from the thermohygrometer and the LI-7200 gas analyser

The water vapour molar density calculated from the thermohygrometer output showed to be a smoothed version of the water vapour molar density directly measured by the LI-7200 gas analyser, as shown for a time period of one hour for the agroforestry plot of Dornburg in Fig. 11. The low-frequency fluctuations were captured, whereas the high-frequency fluctuations were attenuated. A linear regression analysis between both water vapour molar densities yielded a $R^2$ value of 0.85 (based on 29419 data points). We interpret the smoothed water vapour molar density calculated by the thermohygrometer set-up as an effect of the longer response time of the thermohygrometer and the limited sampling frequency of 8 Hz. Spectral analysis of the water vapour mole fraction (Sections 3.5.1 and Fig. 9) derived from the thermohygrometer confirmed the attenuation of high frequencies by the thermohygrometer. The water vapour spectra from the thermohygrometer showed a strong deviation from the theoretical -2/3 slope and from the temperature spectrum at frequencies higher than 0.1 Hz. For frequencies lower than 0.1 Hz the water vapour spectra compared well with the temperature spectrum.

The molar density derived from the thermohygrometer was on average about $100\,\mathrm{mmol\,m^{-3}}$ higher than the molar density measured by the LI-7200 gas analyser during the one hour period. A mean value of $606.32\,\mathrm{mmol\,m^{-3}}$ was found for the

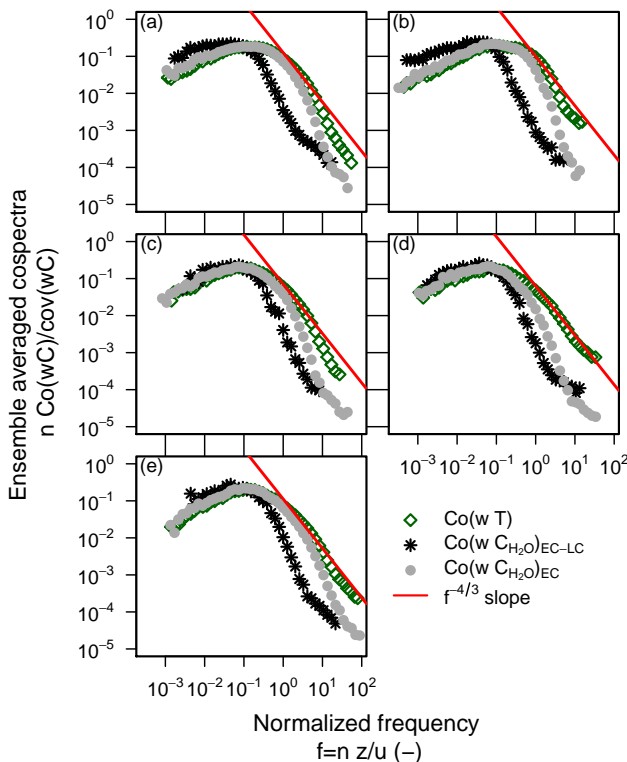

**Figure 10.** Ensemble averaged cospectra of the water vapour flux for the EC- and the EC-LC set-ups (grey and black dots, resp.) and the cospectrum of the sensible heat flux (green dots) versus the normalized frequency over the entire campaign period for Dornburg agroforestry, (a), Dornburg monoculture, (b), Forst agroforestry, (c), Wendhausen agroforestry, (d), and Reiffenhausen agroforestry, (e). Cospectra shown correspond to an unstable stratified atmosphere according to a Monin-Obukhov length between $-650 < L < 0$. Cospectra were filtered for low quality data, corresponding to a flag of 2 following the procedure of Mauder and Foken (2011a) and according to spike removal methods described in Vickers and Mahrt (1997).

thermohygrometer and $514.8\,\mathrm{mmol\,m^{-3}}$ for the LI-7200 gas analyser. We interpret the higher water vapour density derived from temperature, relative humidity and air pressure measurements from the thermohygrometer as an effect of the temperature measurements from the thermohygrometer. We found a $5\,°\mathrm{C}$ higher air temperature from the thermohygrometer compared to the sonic temperature under clear sky condition. The temperature difference is caused by a radiation effect originating from the
5   PVC housing.

In addition, the temperature measurements from the thermohygrometer were attenuated compared to the sonic temperature. We interpret this as an inertia effect of the thermohygrometer. So, if the thermohygrometer complex has a higher thermal mass than the ambient air, the temperature measurements taken by the thermohygrometer are attenuated in the high-frequency range. As the attenuation effect was not found in the relative humidity measurements, we assume that the relative humidity
10   measurements were independent of temperature measurements and therefore relative humidity was not attenuated in the same

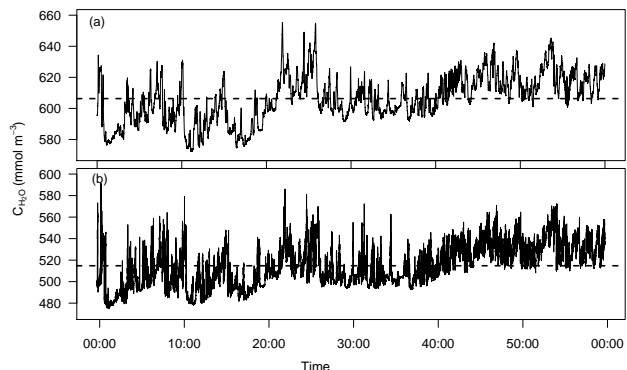

**Figure 11.** Water vapour molar density time series (solid line) and mean (dashed line) for the thermohygrometer, (a), and the LI-7200 gas analyser, (b), at the Dornburg agroforestry plot. The time series represent a 1-hour period from 14:00 to 15:00 hours on 19-July 2016.

way as air temperature. Subsequently, relative humidity fluctuations were conserved and could be used for the calculations of the water vapour mole fraction. In general, the deviation from the mean is of higher interest than the mean itself for the EC method (Baldocchi, 2014). As long as the relative humidity fluctuations are conserved in the calculations of the water vapour mole fraction, a plausible covariance between the water vapour mole fraction and the vertical velocity can be calculated.

## 3.7 Linear regressions of latent heat fluxes from conventional- and low-cost eddy covariance

Results of a linear regression analysis between evapotranspiration rates obtained by the EC and EC-LC set-ups revealed a dependency of the evapotranspiration rates on the high-frequency cospectral correction method used. Evapotranspiration rates obtained by the EC-LC set-up using the Ibrom et al. (2007) high-frequency cospectral correction underestimated evapotranspiration rates obtained by EC using the high-frequency correction after Ibrom et al. (2007) (always used for the EC set-up) at all sites (Table 2). The largest underestimation was 32 % (Forst agroforestry) and the smallest underestimation was 13 % (Dornburg agroforestry), with a median underestimation of 22 % across all five plots.

In contrast, evapotranspiration estimates obtained by the EC-LC set-up using the Moncrieff et al. (1997) high-frequency cospectral correction revealed an underestimation of evapotranspiration rates by the EC-LC set-up of 14 %, 6 %, 5 % and 1 % for the agroforestry plots of Reiffenhausen, Dornburg, Forst and Wendhausen, respectively, and an overestimation by the EC-LC set-up of 8 % for the monocultural agriculture plot of Dornburg relative to the conventional EC set-up (Table 2 and Fig. 12).

The dependency of the evapotranspiration estimates on the chosen high-frequency cospectral correction method may be caused by the assumptions of each method. The Ibrom et al. (2007) high-frequency correction method was initially developed for a closed-path eddy covariance system, with a tube length of about 50 m. The method described in Ibrom et al. (2007) takes into account the dependency of water vapour concentration measurements on relative humidity effects inside the tube. A

**Table 2.** Major axis linear regression of evapotranspiration from EC-LC versus EC, using two high-frequency correction methods (Ibrom et al. (2007) and Moncrieff et al. (1997)). The slopes include the $\pm\,2.5\,\%$ confidence interval. The root mean square error, RMSE, and the coefficient of determination, $R^2$, are given.

| Correction method | Ibrom et al. (2007) | | | Moncrieff et al. (1997) | | |
| Site | Slope/ Intercept | $R^2$ | RMSE (W m$^{-2}$) | Slope/ Intercept | $R^2$ | RMSE (W m$^{-2}$) |
| --- | --- | --- | --- | --- | --- | --- |
| Dornburg AF | 0.87±0.034/ -9.04 | 0.71 | 36.0 | 0.94±0.036/ -10.87 | 0.71 | 35.13 |
| Dornburg MC | 0.78±0.030/ -4.3 | 0.71 | 50.8 | 1.08±0.027/ -5.12 | 0.86 | 34.31 |
| Forst AF | 0.68±0.026/ -0.45 | 0.93 | 74.9 | 0.95±0.045/ -2.9 | 0.90 | 38.5 |
| Wendhausen AF | 0.78±0.016/ -5.8 | 0.93 | 53.71 | 0.99±0.021/ -6.63 | 0.94 | 33.5 |
| Reiffenhausen AF | 0.85±0.034/ -4.1 | 0.90 | 28.13 | 0.86±0.032/ -4.86 | 0.90 | 29.7 |

low-pass cut-off frequency was estimated for each 30-minute period as a function of ambient relative humidity. At least one month of data are suggested to estimate the low-pass cut-off frequency (LI-COR, 2015).

In contrast, the high-frequency correction method after Moncrieff et al. (1997) is purely analytical and applies a fit of the temperature cospectra measured with the sonic anemometer on the water vapour cospectra. This analytical method can be applied independently of meteorological measurements. Furthermore, the correction after Moncrieff et al. (1997) was recommended for either open-path EC systems or under conditions when the intake tube is short and heated (LI-COR, 2015). From an analysis of the high-frequency transfer function from Moncrieff et al. (1997) and the Lorentzian of the infinite impulse response filter from Ibrom et al. (2007) it is evident that the correction of high-frequency losses is better represented by the high-frequency spectral correction of Moncrieff et al. (1997) (see Fig. 13). The transfer function after Moncrieff et al. (1997) is shifted towards higher frequencies and lower frequencies are conserved. According to the Lorentzian (Ibrom et al., 2007) the filtering properties are more pronounced for Ibrom et al. (2007) and low-frequencies ($<10^{-2}$ Hz) are attenuated. Based on the assumptions and recommendations given in Moncrieff et al. (1997) and LI-COR (2015), we decided to apply the correction of Moncrieff et al. (1997) to our EC-LC set-up.

Currently, the authors of the only known study published by Hill et al. (2017) presents a low-cost EC set-up for measurements of $CO_2$ and water vapour fluxes. The authors compared the low-cost EC set-up with a LI-7500 gas analyser sharing the same Campbell Scientific CSAT3 sonic anemometer. They reported a $6\,\%$ flux magnitude overestimation of the latent heat flux obtained by the low-cost EC system relative to the reference EC set-up.

Flux magnitude differences observed for our low-cost set-up are comparable to flux magnitude differences between conventional EC set-ups observed in a recently published study by Polonik et al. (2019). The authors found average differences between $4\,\%$ and $14\,\%$ between water vapour fluxes obtained by different EC set-ups consisting of three different sonic anemometers and five conventional gas analysers.

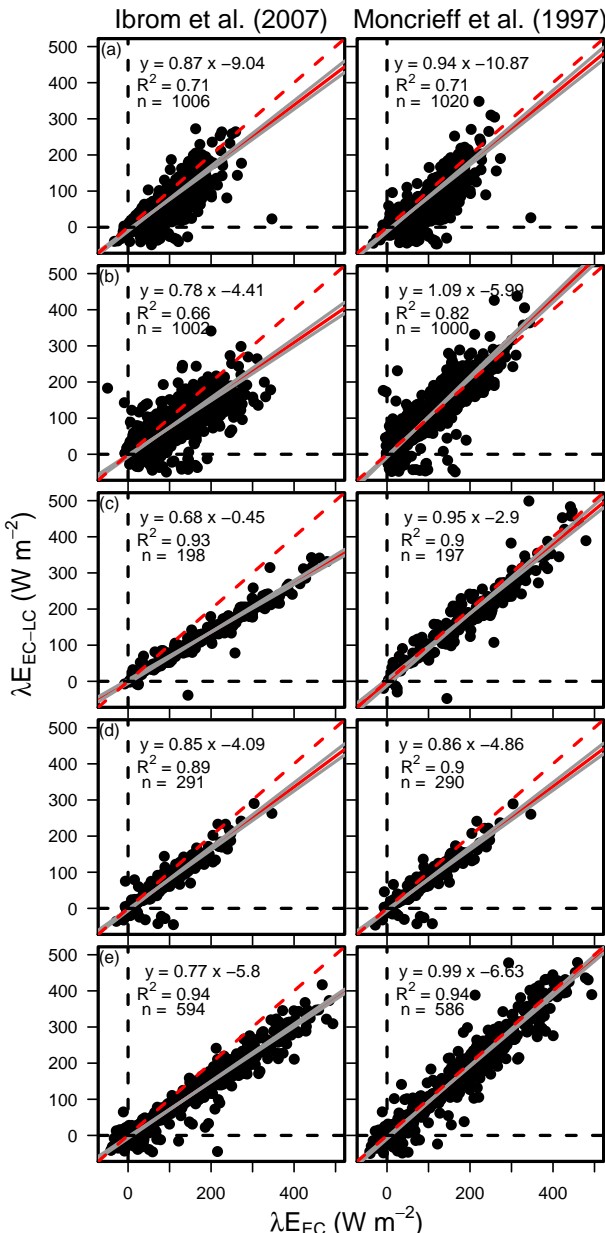

**Figure 12.** Scatter plots of latent heat fluxes obtained by the low-cost EC set-up versus latent heat fluxes obtained by the conventional EC set-up for Dornburg agroforestry, (a), Dornburg monoculture, (b), Forst agroforestry, (c), Reiffenhausen agroforestry, (d), and Wendhausen agroforestry, (e). Latent heat fluxes obtained by the conventional EC set-up were corrected for high-frequency losses by the high-frequency correction method of Ibrom et al. (2007), whereas the latent heat fluxes obtained by the low-cost EC set-up were corrected by, first, the high-frequency correction method of Ibrom et al. (2007) (left site) and, second, the high-frequency correction method of Moncrieff et al. (1997) (right hand site).

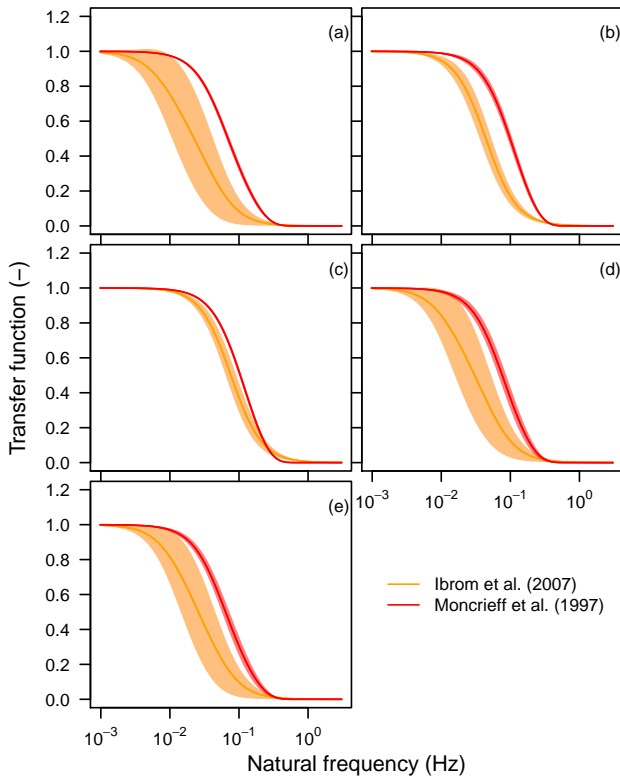

**Figure 13.** Mean and standard deviation of the spectral correction transfer functions vs. the natural frequency for the high-frequency spectral correction methods of Ibrom et al. (2007) and Moncrieff et al. (1997), respectively, for sites, e.g. Dornburg agroforestry, (a), Dornburg monoculture, (b), Forst agroforestry, (c), Reiffenhausen agroforestry, (d), and Wendhausen agroforestry, (e). The transfer function after Ibrom et al. (2007) represent the mean over all infinite impulse response (IIR) filters, approximated by the Lorentzian $H_{IIR}(f|f_c) = \frac{1}{1+(f/f_c)^2}$. $H_{IIR}(f|f_c)$ was estimated for each 30-min period as per the mean ambient relative humidity.

## 3.8 Dependency of the latent heat flux random uncertainty on relative humidity

Common to all sites and both set-ups was a decreasing absolute random uncertainty of the latent heat flux with increasing relative humidity (Fig. 14). At high relative humidity turbulent latent heat fluxes were low, commonly during night time and bad weather conditions. Whereas, during day time and good weather conditions (generally low relative humidity), the fluxes were high. Richardson et al. (2006) described a linear dependency of the absolute random uncertainty on the magnitude of the turbulent fluxes.

For three out of five plots (Dornburg agroforestry and monoculture and Reiffenhausen agroforestry, respectively, Fig. 14 (a), (b) and (e)), we found a lower median random uncertainty for the latent heat fluxes obtained by the conventional EC set-up at low relative humidity, compared to the EC-LC set-up. At high relative humidity ($\geq 70$ %) the median of both random uncertainties was equal.

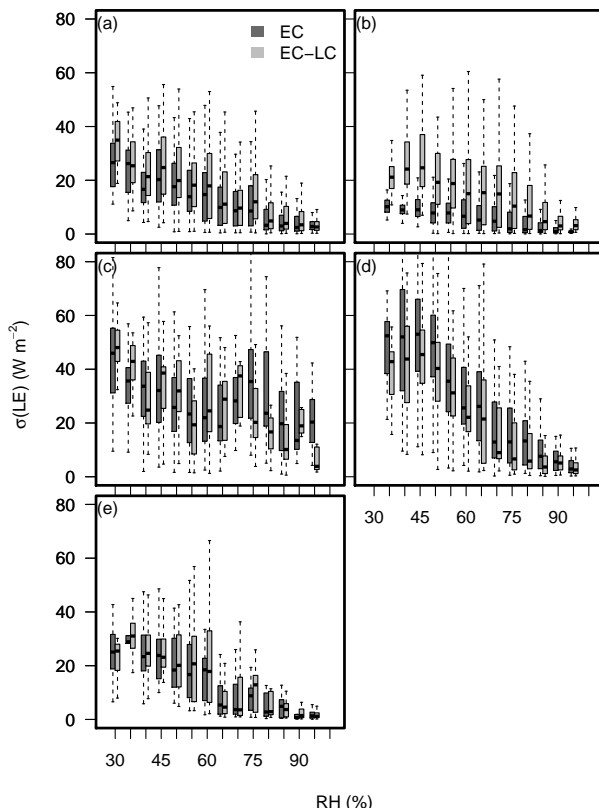

**Figure 14.** Box-whisker plots with random error uncertainty of the latent heat flux calculated by the EC and EC-LC set-up, respectively, versus relative humidity bins of 5 %. Subfigures correspond to plots: Dornburg agroforestry, (a), Dornburg monoculture, (b), Forst agroforestry, (c), Wendhausen agroforestry, (d), and Reiffenhausen agroforestry, (e).

For the other two plots (Fig. 14 (c) and (d)) either a higher or nearly equal mean and standard deviation was found for the latent heat flux random uncertainty from the EC set-up compared to the EC-LC set-up. Furthermore, the standard deviation of the random uncertainty of the latent heat fluxes obtained by the EC and EC-LC set-ups was of the same order of magnitude as their respective mean (Table 3).

## 3.9 Distribution of differences between evapotranspiration estimates

The median of differences between evapotranspiration rates obtained by the EC and EC-LC set-up was negative for the agroforestry plots (Fig. 15 (a), (c), (d) and (e)). This indicates an underestimation of ET rates obtained by the EC-LC set-up, compared to the EC set-up. The distribution of the differences between evapotranspiration rates followed a skewed distribution with a tail towards negative differences of up to $\sim$ -0.15 mm hr$^{-1}$. The tail towards positive values declined sharply after the maximum of the distribution.

**Table 3.** Mean random uncertainties and standard deviations of the latent heat fluxes obtained by the EC and EC-LC set-up.

| Site | $\overline{\sigma(LE_{EC})}$ | $\overline{\sigma(LE_{EC-LC})}$ |
|---|---|---|
| Dornburg AF | $12.94 \pm 15.82$ | $15.76 \pm 16.91$ |
| Dornburg MC | $6.27 \pm 6.01$ | $16.23 \pm 14.42$ |
| Forst AF | $30.87 \pm 18.84$ | $30.84 \pm 18.86$ |
| Wendhausen AF | $27.45 \pm 23.49$ | $23.70 \pm 20.93$ |
| Reiffenhausen AF | $13.2 \pm 14.3$ | $14.4 \pm 15.7$ |

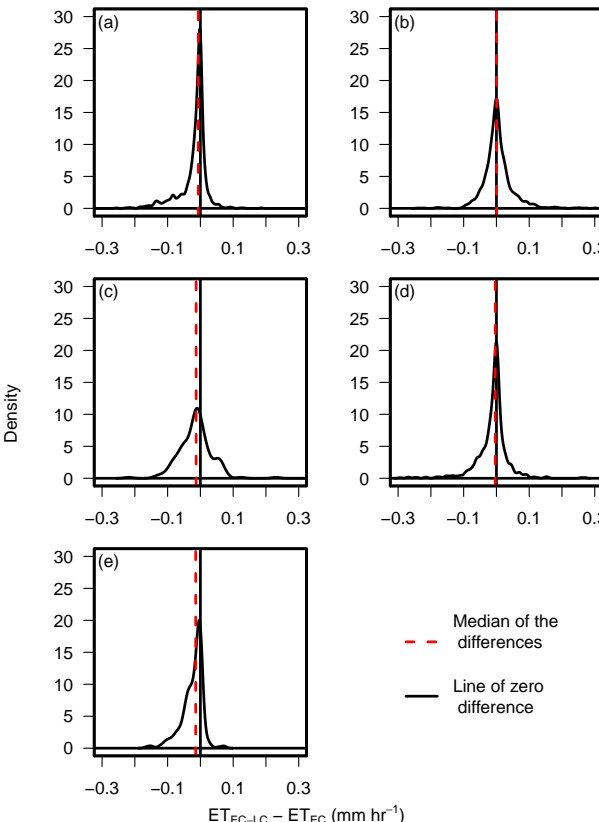

**Figure 15.** Density distribution of differences between evapotranspiration rates obtained by the EC and EC-LC set-up for Dornburg agroforestry, (a), Dornburg monoculture, (b), Forst agroforestry, (c), Wendhausen agroforestry, (d), and Reiffenhausen agroforestry, (e).

At the monocultural agriculture plot at Dornburg (Fig. 15 (b)) there was no significant difference in the median evapotranspiration rates of the two set-ups. The differences were equally distributed towards over- and underestimated ET rates until a zero density of $\pm\,0.1\,\mathrm{mm\,hr^{-1}}$.

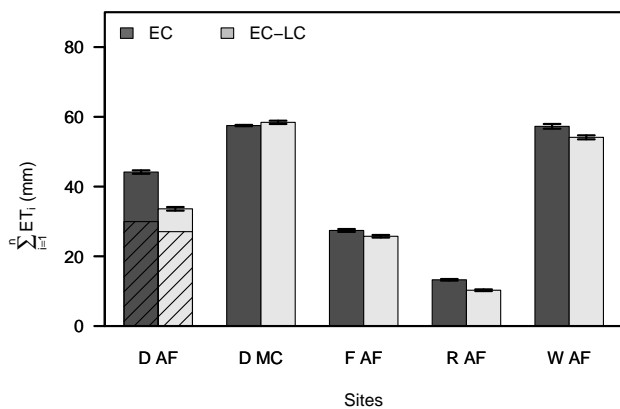

**Figure 16.** Cumulative evapotranspiration rates for the EC and EC-LC set-ups for Dornburg agroforestry, (D AF), Dornburg monoculture, (D MC), Forst agroforestry, (F AF), Wendhausen agroforestry, (W AF), and Reiffenhausen agroforestry, (R AF) over the respective campaign periods (Table A2). The error bars correspond to the summed random uncertainties. The shaded area at Dornburg agroforestry correspond to the cumulative sum of ET filtered for the period of poor performance of the EC-LC set-up. Incomplete records with either of EC or EC-LC missing were omitted.

### 3.10 Cumulative evapotranspiration rates

We observed a lower cumulative evapotranspiration for the EC-LC set-up at all agroforestry plots, relative to the conventional EC set-up (Fig. 16 and 17). In contrast, a higher cumulative ET was found for the EC-LC set-up at the monocultural agriculture plot of Dornburg. The plot of cumulative ET lines in Figure 17 (a I) indicates a discrepancy between the cumulative ET lines at the agroforestry plot of Dornburg. This is caused by a period of poor performance of the low-cost set-up. After removing this period from the data set, we still observed higher ET sums at the AF than at the MC plot, but now differences were comparable to differences observed at the other plots, as indicated by the black and red bars in Figure 16. In general, the observation of underestimated or overestimated (agroforestry vs. monocultural plots) ET rates obtained by the EC-LC set-up relative to the EC set-up is in agreement with the linear regression results presented in Section 3.7.

### 3.11 Annual cumulative ET rates for the agroforestry and the monocultural plot

We wanted to understand how evapotranspiration of agroforestry and monoculture differed. We deployed the EC-LC set-up as a convenient means to obtain continuous long-term evapotranspiration estimates at 30-minute resolution. Here, we present annual cumulative sums of 30-minute evapotranspiration rates for 2016 from all sites, independently of the measuring campaigns.

At the Dornburg site, annual cumulative evapotranspiration rates were higher at the monocultural agriculture plot compared to the agroforestry plot (Fig. 18), which might be caused by the wind-exposed location of the monocultural agriculture plot. The higher wind speed at the monocultural agriculture plot increases the boundary layer conductance and therefore both soil evaporation and plant transpiration increase.

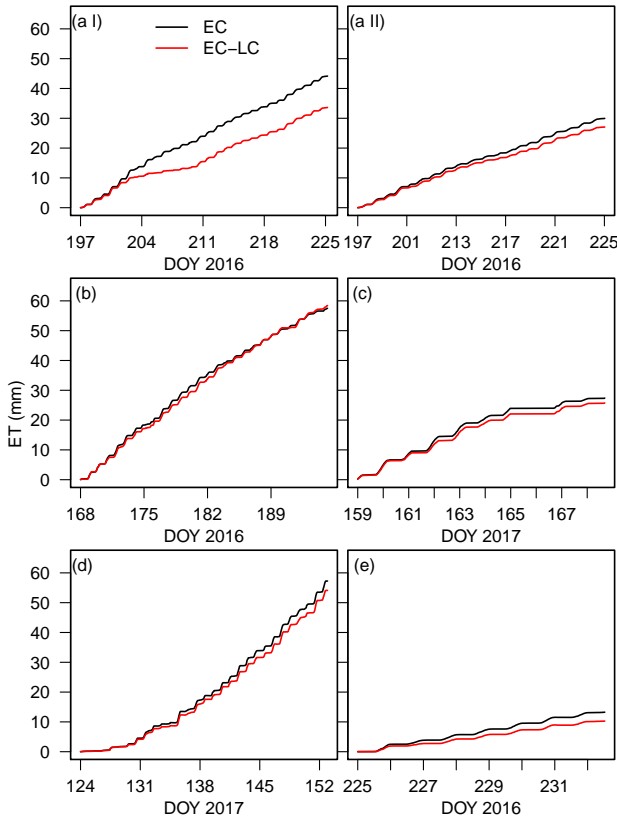

**Figure 17.** 30-minute cumulative evapotranspiration rates for the EC (solid black line) and EC-LC (solid red line) set-ups for Dornburg agroforestry with unfiltered data for the period of poor performance of the EC-LC set-up, (a I), Dornburg agroforestry with filtered data for the period of poor performance of the EC-LC set-up, (a II), Dornburg monoculture, (b), Forst agroforestry, (c), Wendhausen agroforestry, (d), and Reiffenhausen agroforestry, (e), over the respective campaign periods (Table A2). Incomplete records with either of EC or EC-LC missing were omitted.

At the remaining four out of five sites the annual cumulative evapotranspiration rates were higher at the agroforestry plots than at the monocultural agriculture plots (Forst, Wendhausen, Mariensee and Reiffenhausen, Fig. 18). We interpret higher evapotranspiration rates at the agroforestry than at the monocultural plots as an effect of the increased biomass at the agroforestry plot, originating both from the trees and the crops grown between the tree strips. Despite the presence of a leeward side with reduced evapotranspiration caused by the wind reduction and the increased shade, both crops and trees are affected by wind on the windward site. More turbulent conditions are present at the agroforestry plots as caused by the presence of the tree strips, which is indicated by a higher mean roughness length at the agroforestry plots compared to the conventional agriculture plots as shown in Fig. A1 for all sites.

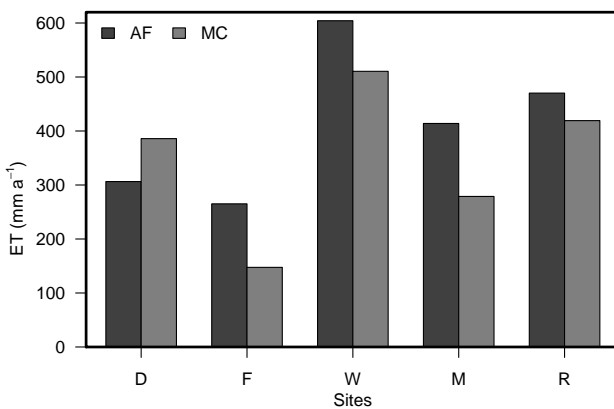

**Figure 18.** Cumulative evapotranspiration rates obtained by the EC-LC set-up at sites Dornburg, (D), Forst, (F), Wendhausen, (W), Mariensee, (M), and Reiffenhausen, (R), for 2016. Incomplete records with either of agroforestry or monoculture missing were omitted. Gap-filling was performed by multiplying the summed ET with the ratio of the number of maximum possible records to the number of missing records.

## 4 Conclusions

We presented a new low-cost eddy covariance set-up, which is comprised of a conventional ultrasonic anemometer and a low-cost thermohygrometer. We applied the eddy covariance method on the vertical velocity component and the water vapour mole fraction derived from the thermohygrometer. The advantages of the set-up are low material costs and low power consumption.

The performance of the EC-LC set-up was comparable to the EC set-up with regards to mean evapotranspiration rates. The set-up specific differences in mean evapotranspiration rates were insignificant compared to the variability between sites.

In detail, we were able to explain more than 80 % of the variability in evapotranspiration obtained by the conventional eddy covariance set-up by the variability of the low-cost eddy covariance set-up. The low-cost eddy covariance set-up is a good alternative to the conventional EC set-up for both conventional agriculture systems and agroforestry ecosystems at a temporal

resolution of 30 minutes.

We showed that under conditions of high relative humidity and low air temperature the flux random error uncertainty of both set-ups was highest. ET rates obtained by the EC-LC set-up with limited frequency response had a lower relative difference to ET rates obtained by the EC set-up at the 10 m measurement height (AF) than at the 3.5 m height given a larger contribution of low-frequency eddies at the larger measurement height.

We anticipate potential applications of the EC-LC set-up in experiments comparing different treatments (management effects, different agriculture systems, water use) and chronosequences after fires or clear cuts. The set-up provides a tool for replicated ET measurements across different ecosystems. With low-cost instruments, flux measurements at existing flux networks such as FLUXNET, ICOS or NEON can be complemented and can be provided at remote and so far underrepresented sites.

*Data availability.* All data used for the figures presented here are provided in the Supplement.

*Author contributions.* C. Markwitz designed and performed the field work, analyzed the data and has written the current manuscript. Dr. L. Siebicke wrote the project scientific proposal, acquired the funding as part of the BonaRes SIGNAL consortium, and contributed to field work, analysis and manuscript writing.

5   *Competing interests.* The authors declare that they have no conflict of interest.

*Acknowledgements.* We kindly acknowledge the funding from the German Federal Ministry of Education and Research (BMBF, project BonaRes, Modul A: Signal 031A562A) and from the Deutsche Forschungsgemeinschaft (INST 186/1118-1 FUGG). We further wish to acknowledge contributions by A. Knohl and M. Herbst to the BonaRes SIGNAL proposal and project design as well as the technical support of field work received by F. Tiedemann, E. Tunsch, D. Fellert, M. Lindenberg, J. Peters (bioclimatology group) and D. Böttger (soil science
10  group of tropical and subtropical ecosystems) from the University of Göttingen.

## Appendix A

**Table A1.** Site locations, agroforestry geometry and stand characteristics.

| Site | Coordinates | No. of tree alleys | System size [m$^2$] | Relative tree cover | Tree height [m] |
|------|-------------|--------------------|--------------------|--------------------|-----------------|
| Reiffenhausen | 51°24'N 9°59'E | 3 | 18700 | 72% | 4.73±0.32 (n=69) |
| Mariensee | 52°34'N 9°28'E | 3 | 69260 | 6% | 4.01±0.33 (n=96) |
| Wendhausen | 52°20'N 10°38'E | 6 | 179738 | 11.52% | 6.21±0.4 (n=114) |
| Forst | 51°47'N 14°38'E | 7 | 391300 | 12% | 6.5±1.8 (n=161) |
| Dornburg | 51°47'N 11°39'E | 7 | 508723 | 8% | 6.4±0.64 (n=160) |

**Table A2.** Temporal extend of the EC measurement campaigns.

| Site | Campaign period |
|------|-----------------|
| Dornburg Conv | 16-June to 14-July 2016 |
| Donburg AF | 14-July to 12-August 2016 |
| Reiffenhausen AF | 12-August to 14-September 2016 |
| Wendhausen | 03-May to 02-June 2017 |
| Forst | 08-June to 08-July 2017 |
| Mariensee | 21-July to 19-September 2017 |

**Table A3.** Instrument separation of the gas analyser relative to the centre of the sonic anemometer into the North, East and vertical direction.

| Site | North [cm] | East [cm] | Vertical [cm] | Year |
|---|---|---|---|---|
| Dornburg MC | 6 | 14 | -21 | 2016 |
| Dornburg AF | -27 | 4 | -26 | 2016 |
| Reiffenhausen AF | 1 | 9 | -20 | 2016 |
| Wendhausen AF | -10 | 0 | -20 | 2017 |
| Forst AF | -12 | 0 | -22 | 2017 |

**Table A4.** Mean air temperature, T, vapour pressure deficit, VPD, global radiation, $R_G$ and the cumulative precipitation, Rain, for the respective site and measurement period.

| Site | T (°C) | VPD (hPa) | $R_G$ (W m$^{-2}$) | Rain (mm) |
|---|---|---|---|---|
| Dornburg MC | 18.6 | 7.35 | 212.6 | 2.1 |
| Dornburg AF | 19.0 | 6.41 | 200.7 | 57.1 |
| Reiffenhausen AF | 19.31 | 8.02 | 219.1 | 26.3 |
| Wendhausen AF | 16.6 | 5.4 | 235.0 | 48.6 |
| Forst AF | 21.4 | 12.02 | 358.8 | 18.9 |

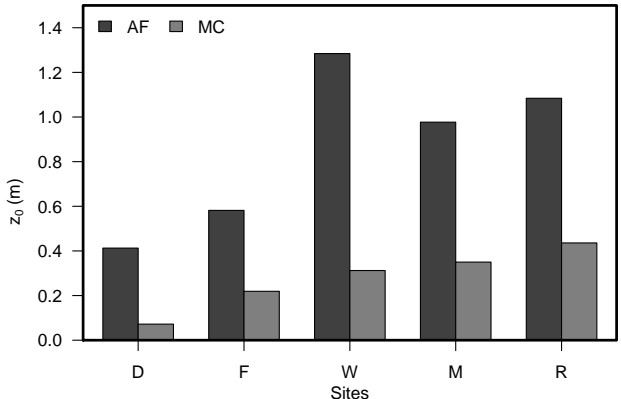

**Figure A1.** Mean roughness length at sites Dornburg, (D), Forst, (F), Wendhausen, (W), Mariensee, (M), and Reiffenhausen, (R), for 2016.

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
