# Peer review of "Low-cost eddy covariance: a case study of evapotranspiration over agroforestry in Germany"

_Atmospheric Measurement Techniques, 2018_

## Referee Comment (RC1) · Timothy Hill (Referee) · 18 Apr 2019

This manuscript provides an interesting approach to low cost ET measurements that have been tested at large number of sites and is a useful addition to the literature. The instrumental approaches described are shown to be effective in comparisons with the LI-7200 systems. The comparison of cumulative ET (Figure 11) is impressive – it would be informative to show cumulative ET lines (perhaps in appendix) to illustrate if the seasonal responses are comparable. Furthermore it would be worth a look in the literature to put in context the size of the differences (are they close to the disagreement between conventional systems).

[Figure]

My first main comment is that I would please like to see are details on: 1) the cost (since this is a low cost system, how low cost is it?); 2) power usage; 3) construction (details needed for people to replicate the build), and 4) maintenance of the low cost system. I see these details as extremely valuable for any readers to replicate this study.

The second main comment I have is that it would be very informative to see details about the actual frequency response of the low cost sensors (RH and T) and if there are environmental dependencies on these response times. It would be good to see a comparison of the sensor specification and actual response times derived from the spectral analyses. A related point is, what was the size of the frequency response correction?

My third main query is what did the energy balance closures look like? ALthough an incomplete assessment of the ET, it would be informative to know the closure for the systems and sites.

Further minor comments are:

Abstract: - A (pedantic) comment on the assumption that Eddy Covariance is appropriate for homogeneous land surfaces: Whilst arguably true (depending on the errors associate with EC) the assumption of homogeneity first needs to be tested using a suitable experimental design. See Hurlbert 1984 (Pseudoreplication and the Design of Ecological Field Experiments). Otherwise our implicit assumption is that the (non-flux) data we have about the full extent of the terrain (which might be limited to little more than a visual/reflectance based observations) is sufficient to predict the fluxes (or at least the variability - or lack of - in fluxes) – and if this is the case why use EC?

-Line 8: Given the general lack of energy balance closure for the EC method, I don't think the 'true' ET flux is known. Therefore, 'underestimation' and 'overestimation' are more accurately termed 'underestimation relative to the conventional system'.

Page 3: Can you describe the site fetch? What are the heights of the trees and the

crops? Reiffenhausen is a small site 18,700 m2 ($\sim$1.9 ha), what is beyond the extent of this site (and likely in your flux footprint)?

Discussion: - I am reluctant to recommend citing my own paper, but as it is one of the only other studies to calculate ET from a low cost RH sensor, I think comparisons with the LE fluxes/approach from Hill GCB 2017 (and any others) should be made somewhere in the discussion.

-Page 6 It would be useful to know the indicative cost and power usage for both systems. What is the volume of the thermohygrometer housing? What is the form of the housing? What response time (and measurement principle) did the temperature sensor of the BME280 use?

- Page 6: it is not entirely clear to me if the systems shared the same sonic, and if not, what was the spatial separation of the comparison system?

-Page 7: I am interested in how much data was filtered through QC and how you filtered data for the LC system?

-Page 8: It would be useful to know the time response of the temperature sensor. Figure B1 does not give a good insight into this response as it convolves: sensor response; sensor noise; housing attenuation and variability of scalar (i.e. RH or T). A look at the spectra/cospectra of the sensors (and a modelled attenuation of the sonic-T would give a much clearer idea (and quantification) of the total combined attenuation of the sensor and housing.

-page 9: provide details here, or later on about the timelag. Are you sure this is due to the vertical separation? (if so it should be dependent on W). Alternatively it could be due to the sensor response/processing time and therefore it reasonable to expect it may include a T/RH dependency.

-page 15: Fig6 It is interesting to see that the LI-7200 is highly attenuated and more sensitive to RH than the LC system. Indeed attenuation of the LI-7200 in panel c (and

[Figure]

even more so in d) is significant and indicates a very poor frequency response for this system. Any thoughts on why? Did you run with filters and did they clog frequently?

Fig 6, can you please clarify (as I assume that the RH is specific for the LI-7200 and the LC sensor (with its higher temperatures and presumably lower RH). Either way the comparison is complicated: if ambient RH is used, then the sensors are effectively seeing different RH, alternatively if sensor RH is used, then the spectra contain different data (i.e. wind speed/stability might differ). Neither point are likely to be particularly significant to the overall interpretation, but should be clarified.

Fig 6/7: please include the criteria for data shown, what correlation strength/LE/stability classes are included?

-Page 17: The linear regressions are very important and it would be very useful to see the scatter plots associated with these to see if they are well behaved. -page 21: figure 12. It is not clear how the 2016 annual ET fluxes were arrived at given the campaign basis of the measurements. Table A3 implies some sites were not measured in 2016.

---

## Referee Comment (RC2) · Anonymous Referee #3 · 10 May 2019

**General comments**

This manuscript presents a test of a low-cost hygrometer manufactured by Bosch GmbH being used for eddy-covariance measurements. The sonic anemometer is the same as for regular eddy-covariance system being deployed. Another difference between the low-cost system and the regular system is the data acquisition, which is realized by a Raspberry Pi instead of a Campbell CR6 data logger. The regular EC system has a Licor LI7200 for measuring water vapor and $CO_2$ fluctuations. I doubt that the data acquisition causes significant differences in the collected data since both systems are recording digitally. So, the main question of this study is, whether the

precision and the spectral response characteristics of the Bosch hygrometer are sufficient for eddy covariance applications. The results of evapotranspiration show a good agreement, if adequate spectral corrections are applied, which leads the authors to the main conclusion that this low-cost system is an alternative when a larger number of measurement units is required for a certain application. I generally agree with this assessment; however, I suggest that a more extensive evaluation of the spectral response characteristics of the Bosch sensor based on the collected field data should be presented, e.g. the system's cut-off frequency based on in-situ assessment method of Ibrom et al. (2007) and the transfer function of the Moncrieff et al. method. This would perhaps also better explain why the one method gave different results than the other.

Minor comments

Abstract: I find the abstract too long, I am not sure though, if this journal has any limits in that respect. E.g. the introductory sentences could be shortened. Nevertheless, I would suggest to mention the main results, perhaps even including information about the RMSE.

P2, L10-21: It is not clear how this is relevant for the topic of this paper. Perhaps omit these sentences, although they are correct.

L9, L7: How were the clocks of the two systems synchronized and how good was this synchronization. It needs to be better than 0.05 s.

P10, L17: Since you analyzed the spectra already, I suggest that you also empirically determine and present the cut-off frequency of the Bosch sensor, also in order to verify the response time provided in the specifications.

---

## Author Comment (AC1) · 7 Jun 2019

We thank you for reviewing our manuscript.

Please find attached the detailed author response and a document with tracked changes in the supplement.

Please also note the supplement to this comment: https://www.atmos-meas-tech-discuss.net/amt-2018-392/amt-2018-392-AC1-supplement.zip

---

## Author Comment (AC2) · 7 Jun 2019

We thank you for reviewing our manuscript.

Please find attached the detailed author response and a track changes document in the supplement.

Please also note the supplement to this comment:
https://www.atmos-meas-tech-discuss.net/amt-2018-392/amt-2018-392-AC2-supplement.zip

---

## Author Response (AR1)

**Author response to the reviewers comment from Timothy Hill on the manuscript amt-2018-392: "Low-cost eddy covariance: a case study of evapotranspiration over agroforestry in Germany"**

We thank you for your feedback, suggestions and helpful comments on the manuscript. In the current document we give a point-by-point answer on above mentioned referee report. We show first the referee comments (**RC**) and secondly the answer of the authors (**AR**). Changes made in the manuscript can be found in the track changes document attached to the current document. Figure numbers and references refer to the track-changes document, if not otherwise stated.

**1. RC:** This manuscript provides an interesting approach to low cost ET measurements that have been tested at large number of sites and is a useful addition to the literature. The instrumental approaches described are shown to be effective in comparisons with the LI-7200 systems. The comparison of cumulative ET (Figure 11) is impressive – it would be informative to show cumulative ET lines (perhaps in appendix) to illustrate if the seasonal responses are comparable. Furthermore it would be worth a look in the literature to put in context the size of the differences (are they close to the disagreement between conventional systems).

**1. AR:** Figure 16 shows the cumulative sum of half-hourly evapotranspiration rates for the respective campaign times of approximately four weeks duration. The data were filtered for implausible values and gaps were not filled for this analysis to reduce the inferred error caused by gap-filling. We included the cumulative ET lines for the respective campaign periods in Figure 17. The figure points out that both set-ups recover properly the temporal changes of evapotranspiration during the campaign periods, caused by the plant physiological response of the underlying ecosystem to changes in meteorological driver such as incident radiation, air temperature and the vapour pressure deficit. The difference between both set-ups at the Dornburg AF site was caused by a period of bad performance of the low-cost system. If the period was discarded from the data, the difference between EC and EC-LC at the Dornburg AF site was comparable to differences at the other sites, as shown in Figure 16. We included figures 16 and 17 as shown in the current document. Regarding the comparison of differences found for the low-cost set-up with conventional systems, we included some literature (including your publication) in Section 3.7 of the manuscript.

15 Currently, the authors of the only known study published by Hill et al. (2017) presents a low-cost EC set-up for measurements of CO2 and water vapour fluxes. The authors compared the low-cost EC set-up with a LI-7500 gas analyser sharing the same Campbell Scientific CSAT3
 20 sonic anemometer. They reported a 6% flux magnitude overestimation of the latent heat flux obtained by the low-cost EC system relative to the reference EC set-up.

Flux magnitude differences observed for our low-cost set-up are comparable to flux magnitude differences between 25 conventional EC set-ups observed in a recently published study by Polonik et al. (2019). The authors found average differences between 4% and 14% between water vapour fluxes obtained by different EC set-ups consisting of three different sonic anemometers and five conventional gas 20 analysers.

Figure 16. Cumulative evapotranspiration rates for the EC and EC-LC set-ups for Dornburg agroforestry, (D AF), Dornburg monoculture, (D MC), Forst agroforestry, (F AF)and-, Wendhausen agroforestry, (W AF), and Reiffenhausen agroforestry, (R AF) over the respective campaign periods (Table A2). The error bars correspond to the summed random uncertainties, which were added. The shaded area at Dornburg agroforestry correspond to the cumulative evapotranspiration rates un of ET filtered for the period of poor performance of the EC-LC set-up. Incomplete records with either of EC or EC-LC missing were omitted.

Figure 17. 30-minute cumulative evapotranspiration rates for the EC (solid black line) and EC-LC (solid red line) set-ups for Dornburg agroforestry with unfiltered data for the period of poor performance of the EC-LC set-up, (a I), Dornburg agroforestry with filtered data for the period of poor performance of the EC-LC set-up, (a II), Dornburg monoculture, (b), Forst agroforestry, (c), Wendhausen agroforestry, (d), and Reiffenhausen agroforestry, (e), over the respective campaign periods (Table A2). Incomplete records with either of EC or EC-LC missing were omitted.

**3.10 Cumulative evapotranspiration rates**

We observed a lower cumulative evapotranspiration for the EC-LC set-up at all agroforestry plots, compared-relative to the conventional EC set-up (Fig. 16 and 17). In contrast, a higher cumulative ET was found for the EC-LC set-up at the monocultural agriculture plot of Dornburg. The plot of cumulative ET lines in Figure 17 (a I) indicates a discrepancy between the cumulative ET lines at the agroforestry plot of Dornburg. This is caused by a period of poor performance of the low-cost set-up. After removing this period from the data set, we still observed higher ET sums at the AF than at the MC plot, but now differences were comparable to differences observed at the other plots, as indicated by the black and red bars in Figure 16. In general, the observation of underestimated or overestimated (agroforestry vs. monocultural plots) ET rates obtained by the EC-LC compared-set-up relative to the EC set-up are-is in agreement with the linear regression results presented in Section 3.7.

**2. RC:** *My* first main comment is that I would please like to see are details on: 1) the cost (since this is a low cost system, how low cost is it?); 2) power usage; 3) construction (details needed for people to replicate the build), and 4) maintenance of the low cost system. I see these details as extremely valuable for any readers to replicate this study.

**2. AR:** We included more required information in the section "Instrumental set-up - Low-cost eddy-covariance (EC-LC) installation".

Changes in the manuscript:

**2.2.3 Low-cost eddy-covariance (EC-LC) installation**

The low-cost eddy-covariance set-up comprised of shared the same ultrasonic anemometer (uSONIC3-omni) as used for the conventional EC method and a set-up. The water vapour mole fraction was derived from the combined digital pressure, relative humidity and air temperature sensor (BME280 ,-manufactured by Robert Bosch GmbH, Stuttgart, Germany) ((hereafter named thermohygrometer, Fig. 2 depicts the low-cost set-up). The measuring principle is resistive, capacitive and based on diode voltage measurements for the air pressure, humidity and temperature sensor, respectively. The ultrasonic anemometer measured the three-dimensional wind speed and the ultrasonic temperature at a frequency of 20 Hz, whereas the thermohygrometer measured the air temperature, relative humidity and air pressure at a sampling frequency of 8 Hz. The specified response time of the thermohygrometer for relative humidity measurements is 1 s to overcome 63 % of a step change from 90 % to 0 % or 0 % to 90 % relative humidity -at 25°C air temperature.

The thermohygrometer was placed 0.5 m below the centre of the sonic anemometer in a PVC housing to protect the thermohygrometer from precipitation. A-The PVC housing consisted of an outer and an inner cylinder. The inner cylinder was perforated on the top to provide a continuous air flow of 15 lpm, generated by a ventilator (HA30101V3-0000-A99, *Sunonwealth Electric Machine Industry Co. Ltd.*, Fresnes Cedex, France). The ventilator was placed below the thermohygrometer provided a continuous air flow of 15 lpm. inside the inner cylinder. The volume of the inner cylinder 25 was 98.1 cm3.

The absolute accuracy tolerance of the thermohygrometer relative humidity sensor was specified as  $\pm 3 \%$  relative humidity (Bosch Sensortee GmbH, 2016). Data for 20 to 80% relative humidity at 25°C, for the temperature sensor an absolute accuracy tolerance of  $\pm 0.5^{\circ}C$  at 25°C and  $\pm 1 °C$ for a temperature range of 0 to 65°C was specified and for the pressure sensor an absolute accuracy tolerance of  $\pm 1$  hPa (300-1100 hPa, 0-65°C) (Bosch Sensortec GmbH, 2016).

Digital data from the thermohygrometer were recorded 35 via the i2c protocol and stored on a RaspberryPi model B+ (*Raspberry Pi Foundation*, Cambridge, UK). The potential of the low-cost EC set-up are replicated measurements of evapotranspiration across different ecosystems. The relative cost of the low-cost set-up (featuring a sonic anemometer, a RaspberryPi and the thermohygrometer of low cost) is about 8-10 % of a conventional EC set-up.

The thermohygrometer points out with very low power consumption of approximately 3.6  $\mu A$  at a sampling frequency of 1 Hz (9.4e-5 W at 8 Hz, powered with 3.3 45 V and if all three variables are measured simultaneously) and the RaspberryPi has a maximum power consumption

of about 1.1 W if all three variables are measured at the same time. The set-up requires low maintenance. The sensors needs to be properly installed, such as they are protected

against precipitation. Furthermore, a stable power supply is required. Currently, two out of ten sensors were deployed for a duration of two years.

**3. RC:** The second main comment I have is that it would be very informative to see details about the actual frequency response of the low cost sensors (RH and T) and if there are environmental dependencies on these response times. It would be good to see a comparison of the sensor specification and actual response times derived from the spectral analyses. A related point is, what was the size of the frequency response correction?

**3. AR:** In the following we want to address the spectral response characteristics of the BME280 thermohygrometer in two ways, first, in terms of the cut-off frequency and as the derived sensor time constant and, second, in terms of the spectral correction factor for water vapour.

**Changes in the manuscript:**

**1. Cut-off frequency and sensor time constant**

We included a new section (Section 3.4: Sensor cut-off frequency and time constant) on the sensors cut-off frequency and time constant into the manuscript and showed the dependency of the time constant on relative humidity (Figure 8).

**3.4 Sensor cut-off frequency and time constant**

The nominal time response of the relative humidity sensor as part of the thermohygrometer yields a theoretical sensor

- se cut-off frequency of 0.16 Hz (6.3 s) calculated from Eq. 13. Under field conditions we observed a mean cut-off frequency of  $0.063 \pm 0.02 \text{ Hz}$  for the low-cost thermohygrometer and  $0.3 \pm 0.2 \text{ Hz}$  for the LI-7200 gas analyser across five plots and all humidity classes (from 30 %
- to 90% relative humidity bins). The respective mean time constant was  $2.8 \pm 1$  s for the low-cost thermohygrometer and  $0.6 \pm 0.3$  s for the LI-7200 gas analyser (see Fig. 8). For both sensors we found an exponential increase of the time constant with relative humidity (see Fig. 8).
- 45 Under field conditions, the cut-off frequency and the respective time constant of the thermohygrometer were inferior to the one given in the specifications. We interpret this as caused by the design of the enclosure. The thermohygrometer is placed at the end of a cylinder with
- 50 the ventilator directly below, so that the flow velocity is decelerated. Subsequently, the decelerated flow velocity leads to a limited signal response. One suggestion for improvement of the frequency response would be to place

the thermohygrometer inside a longer tube with a freely moving air stream. This ensures a faster air exchange inside the measurement cell of the thermohygrometer and hence a faster response time.

. . . . . .

Figure 8. Time constant against relative humidity for the LI-7200 (black solid lines) and the thermohygrometer (red solid lines). Dashed lines with the same colour coding as for data shown and values written, correspond to the mean time constant for the respective sensors across all relative humidity classes. Sites correspond to Dornburg agroforestry, (a), Dornburg monoculture, (b), Forst agroforestry, (c), Reiffenhausen agroforestry, (d), and Wendhausen agroforestry, (e).

**2. Spectral correction factor for water vapour**

| Site             | Spectral correction
factor (-) |                    | Spectral correction factor flux magnitude change (%) |                   |
|------------------|-----------------------------------|--------------------|------------------------------------------------------|-------------------|
| Method           | EC EC-LC                          |                    | EC                                                   | EC-LC             |
| Dornburg AF      | 1.11                              | 1.76               | 6.9                                                  | 40.82             |
| Dornburg MC      | 1.21                              | 3.01               | 14.3                                                 | 60.9              |
| Forst AF         | 1.1                               | 1.99               | 9.9                                                  | 47.7              |
| Reiffenhausen AF | 1.11                              | 1.31               | 9.4                                                  | 42.3              |
| Wendhausen AF    | 1.16                              | 1.74               | 5.9                                                  | 21.83             |
| Mean+-sd         | 1.14 ±0.05                 | 1.962 ±0.64 | 9.28 ±3.3                                     | 42.7 ±14.1 |

**Table 1:** Median spectral correction factor and the impact of the spectral correction factor on the flux magnitude change.

We found a higher frequency correction factor for water fluxes (combines the correction for high and low-frequency losses) obtained by the EC-LC set-up than for the EC set-up with a median flux increase of 97.4% and 14.6% (see Table 1 and Figure 6 a), respectively.

The effect of the spectral corrections on a flux magnitude increase was most pronounced for the low-cost set-up than for the conventional EC set-up with an overall flux magnitude increase of 42.7  $\pm 14.1$  % and 9.28  $\pm 3.3$  % for the EC-LC and the EC set-up, respectively (see Figure 3 and Table 1 of the current document).

We found the highest median spectral correction factor (3.01) and the highest flux magnitude increase (60.9%) caused by the high-frequency correction for the low-cost setup of the monocultural agriculture plot of Dornburg. We interpret the higher spectral correction factor as caused by different measurement heights, with a measurement height of 3.5 m at the monocultural agriculture plot of Dornburg and a measurement height of 10 m at the agroforestry plot of Dornburg. At the lower tower high frequency eddies are more likely than at the taller tower. As the nominal time response (1 s) given in the specifications and the estimated time response are quite low, the flux loss is high and needs to be corrected for.

We included information on the spectral correction factor into Section 3.3 ("Effect of spectral- and WPL corrections on evapotranspiration rates from low-cost eddy covariance") and Figure 7 into the manuscript.

The overall impact of spectral corrections on a change of the turbulent latent heat fluxes was stronger for the EC-LC set-up compared to the EC set-up. Here, we quantify the overall impact of spectral corrections on latent heat fluxes in terms of the spectral correction factor (SCF) calculated for each 30-minute period. The 30-minute SCF was multiplied with the respective uncorrected flux. A SCF larger than one indicates a flux magnitude increase, whereas a SCF lower than one indicates a flux magnitude decrease. Box-whisker plots of 30-minute SCFs for each site and each set-up are shown in Figure 7 (a). We found a mean SCF of  $1.96 \pm 0.64$

for the EC-LC set-up and  $1.14 \pm 0.05$  for the EC set-up across all sites, indicating a mean flux magnitude increase of 96% for the EC-LC set-up and a mean flux magnitude increase of 14% for the EC set-up. The mean SCF presented 5 here integrates both night and day time periods. Thus, a

- high SCF during night time with commonly low latent heat fluxes leads to a smaller change of the flux magnitude than during day time, when fluxes are commonly high. Therefore, we also present the sum of 30-minute ET rates corrected
- 10 for spectral losses and the sum of the total ET attributed to the spectral corrections in Figure 7 (b). The part of the total corrected ET attributed to the spectral corrections was higher for the EC-LC set-up compared to the EC-set-up and amounted on average to  $42.7 \pm 14.1$ % of total ET for the
- 15 EC-LC set-up and  $9.3 \pm 3.3$  % of total ET for the EC set-up. Across sites, we found the highest median spectral correction factor of 3.01 and the highest part of the total corrected ET attributed to the spectral corrections of 60.9 % for the EC-LC set-up at the monocultural agriculture plot
- 20 of Dornburg. We interpret this as a measurement height dependency of the spectral corrections. The measurement height at the agroforestry plots was 10 m and at the monocultural agriculture plots the measurement height was 3.5 m. We assume that high-frequency eddies are more likely
- 25 close to the surface. Therefore, a detected turbulent signal at the lower measurement height would be shifted towards high frequencies compared to the detected turbulent signal at the higher measurement height (Aubinet et al., 2012). If a sensor is not capable of detecting the turbulent signal in the bird of the detection of the sensor is not capable of detecting the turbulent signal in the bird of the sensor is not capable of detecting the turbulent signal in the bird of the sensor is not capable of detecting the turbulent signal in the
- w high frequency range of the spectrum, the signal is attenuated and needs to be corrected.